# GADA: Geometry-Aware Deformable Aggregation
# for Image-Based Gaussian Splatting

**Siwoo Lim**[1]  **Sunjae Yoon**[2]  **Gwanhyeong Koo**[1]  **Chang D. Yoo**[1]

## Abstract

Gaussian Splatting has achieved significant improvements by incorporating warping-based techniques. However, such methods suffer from pixel-level inaccuracies due to uncertain geometry. This uncertainty leads to spatial misalignments in the warped images, which disrupt residual learning used in warping-based methods and fundamentally limit the gains of correction, particularly on thin structures and high-frequency details. Driven by our insight that useful visual cues are not lost but locally preserved under slight displacement, we propose Geometry-Aware Deformable Aggregation (GADA). This method introduces an iterative refinement module with deformable offsets to actively correct spatial misalignments and recover these displaced cues. Furthermore, to address the limitations of standard pipelines where visibility checks (i.e., thresholding) often discard valid pixels and multi-view warped image fusion relies on naive mean aggregation, our module is coupled with an implicit confidence weighting mechanism that selectively suppresses unreliable evidence. Consequently, our approach outperforms prior warping-based Gaussian Splatting, preserving high-frequency quality while achieving $2.13\times$ faster FPS. The code is publicly accessible at https://github.com/siw00-lim/GADA

## 1. Introduction

3D Gaussian Splatting (3DGS) (Kerbl et al., 2023) has established itself as the de facto standard for real-time radiance field rendering, significantly outperforming previous coordinate-based methods (Mildenhall et al., 2021). Recent works have further refined 3DGS in terms of anti-aliasing (Yu et al., 2024), geometry (Steiner et al., 2025;

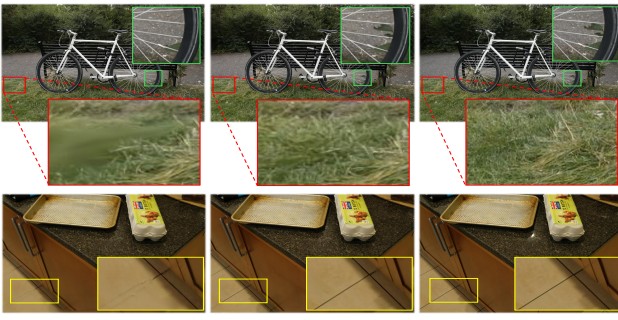

*Figure 1.* **Comparison of detail recovery in challenging regions.** (a) Existing warping-based methods suffer from content blur (e.g., missing foliage details behind the spokes, blurred grass). (b) Our method effectively recovers sharp high-frequency details that closely match the (c) Ground Truth.

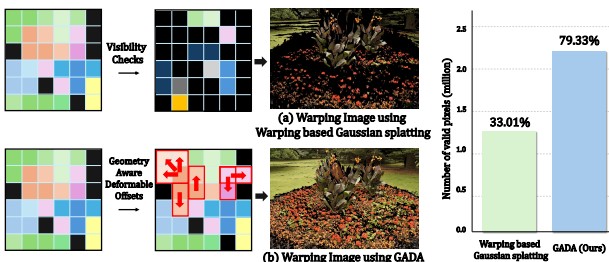

*Figure 2.* **Comparison of warped image processing strategies.** Visibility checks discard valid cues, retaining only 33.01% of pixels. (b) GADA actively corrects misalignments, recovering lost evidence and boosting valid pixel density to 79.33%.

Guédon & Lepetit, 2024), and density control (Ye et al., 2024). However, reconstructing intricate high-frequency details relying solely on explicit 3D primitives remains challenging, often resulting in over-smoothed renderings in complex regions.

To address these limitations, warping based approaches, such as IBGS (Nguyen et al., 2026), have emerged as a powerful alternative. This method utilizes warped images that are reconstructed by mathematically reprojecting pixels from source views[1] into the target view using the scene geometry[2], typically derived from the 3D Gaussians them-

---

[1]Korea Advanced Institute of Science and Technology (KAIST), Republic of Korea [2]Chung-Ang University, Republic of Korea. Correspondence to: Chang D. Yoo <cd_yoo@kaist.ac.kr>.

*Proceedings of the 43[rd] International Conference on Machine Learning*, Seoul, South Korea. PMLR 306, 2026. Copyright 2026 by the author(s).

---

[1]source views denote the set of all training images used for 3D Gaussian Splatting.

[2]Specifically, the depth map rendered from 3D Gaussians and the relative camera extrinsics are employed to perform pixel-wise warping via homography or back-projection.

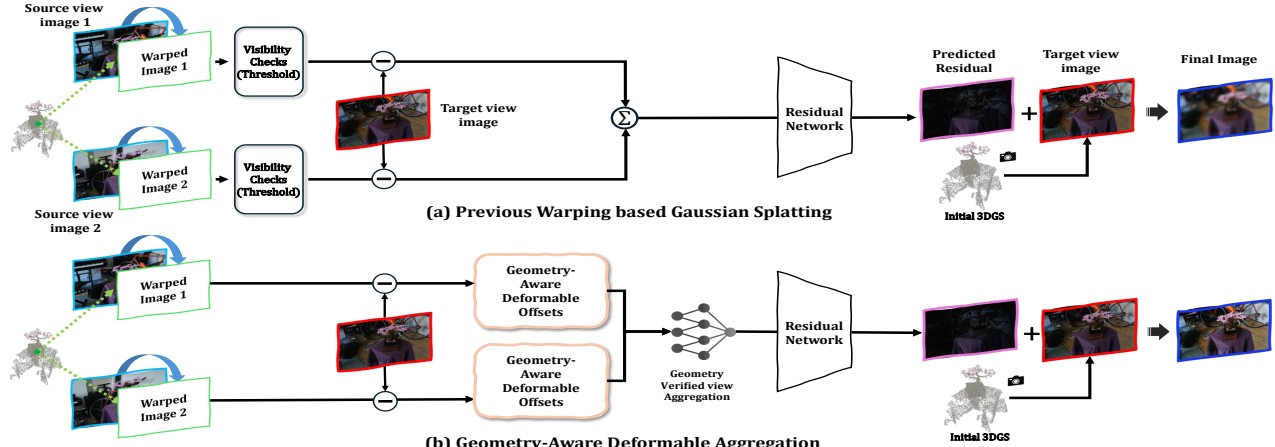

Figure 3. **Comparison of conceptual pipelines between (a) previous warping based Gaussian Splatting and (b) our proposed Geometry-Aware Deformable Aggregation (GADA).** (a) Previous methods rely on visibility checks and a mean aggregation ($\Sigma$), which often fail to handle geometric misalignments, resulting in blurred high-frequency details. (b) To address these, our GADA framework introduces Geometry-Aware Deformable Offsets. These offsets are learned iteratively to compensate for geometric inaccuracies, enabling the flexible alignment of pixels from warped images. By integrating these aligned features through Geometry Verified View Aggregation, our model effectively restores sharp details and ensures robust residual prediction even in regions with complex geometry.

selves as illustrated in Fig. 3 (a). By subtracting the original target view rendering from the warped image, the warping based approach isolates visual cues that represent missing or inaccurately predicted information. These cues serve as essential inputs for the subsequent neural network (referred to as the residual network), enabling more precise prediction of the final residuals, which is and then added to the original target view rendering.

Nevertheless, the efficacy of warping-based techniques is hindered by spatial misalignments arising from the uncertain geometry inherent in 3D Gaussian Splatting. Geometric inaccuracies such as poor depth estimation in thin structures or the presence of floaters inevitably lead to pixel-level discrepancies, which ultimately results in the loss of high-frequency details within the warped images. To mitigate these errors, existing approaches predominantly rely on heuristic visibility checks[3](thresholding) to discard pixels deemed unreliable, followed by a naive mean aggregation, where features derived from each warped image for training of the residual network are simply averaged. This fails to reflect the varying reliability of individual views, as it condenses distinct multi-view information into a single, unweighted representation. However, such strategies fall short of being a comprehensive solution, as these heuristic checks indiscriminately discard even valid visual cues. Specifically, as illustrated in Figure 2 (a), valid pixels within the warped images are suppressed such that the resulting images become excessively sparse and uninformative. Thus, this sparsity deprives the residual network of the high-frequency cues necessary to generate effective residuals. Unfortunately, the current pipeline simply abandons the very evidence required for detailed reconstruction. This fundamentally limits the

potential gains from the warping process; as a result, the residual fails to yield additional improvements in complex regions, remaining blurry as shown in Figure 1.

To overcome these shortcomings and enable robust residuals from warped images, we propose Geometry-Aware Deformable Aggregation (GADA) in Figure 3 (b). GADA does not follow previous assumption that pixels failing visibility checks are merely discardable cues; instead, we redefine them as displaced cues. In a warped image derived from inaccurate geometry, such pixels essentially represent valid visual data that have been mapped to an incorrect location. Therefore, we posit that the true visual cue is not lost but is guaranteed to exist within a bounded local region surrounding the erroneous coordinate. Building on this premise, GADA reformulates the problem from one of discarding pixels due to spatial misalignment to a task of local search and active correction. As illustrated in Figure 2 (b), we employ an Geometry-Aware Deformable Offset to search the local neighborhood and compensate for pixel displacements. This process effectively retrieves valid visual evidence that prior methods would have discarded.

Furthermore, to prevent information dilution from naive mean aggregation, we introduce Geometry-Verified View Aggregation. By assigning adaptive weights to features across warped views, this module prioritizes relevant information for precise residual learning. GADA resolves sparse alignment issues, successfully recovering thin structures and high-frequency details. By eliminating heuristic visibility checks, it also delivers significantly faster rendering than existing warping-based methods.

---

[3]Detailed method of the heuristic visibility checks are provided in Appendix A.1.

## 2. Related Works

### 2.1. Gaussian splatting and quality improvements.

3D Gaussian Splatting (3DGS) renders novel views by alpha blending splatted Gaussians. Follow-up works improve fidelity by addressing aliasing and artifacts: Mip-Splatting mitigates aliasing via principled filtering, and AAA-Gaussians reduces inconsistencies by incorporating faithful 3D evaluation. Others redesign reconstruction kernels to model sharp discontinuities; 3D-HGS introduces half-Gaussian kernels (Li et al., 2025). Complementary efforts refine densification and placement (Rota Bulò et al., 2024; Ye et al., 2024), while 3DGS-MCMC reformulates placement via stochastic sampling (Kheradmand et al., 2024). SuperGaussians increases expressiveness with spatially varying attributes (Xu et al., 2024). Unlike works refining primitives, we target robustness and efficiency in hybrid Gaussian pipelines leveraging warped evidence to recover high-frequency details lost by geometric inaccuracies, resolving ambiguities via learnable offsets and confidence-driven fusion, bypassing brittle heuristics to ensure high-fidelity reconstruction.

### 2.2. Image-Based Rendering and warping based Gaussian Splatting

Image-based rendering (IBR) approaches synthesize novel views by directly utilizing pixels or features from a set of source images, blending corresponding samples based on estimated geometry or correspondence. Early approaches (Buehler et al., 2001) rely on explicit proxy geometry or viewing angles for blending, while later methods improve correspondence via better geometry estimation or optical flow. With the advance of neural rendering, researchers have explored integrating learned feature aggregation into IBR (Yu et al., 2021; Chen et al., 2021), typically employing heavy aggregation networks (e.g., transformers) over epipolar samples (Wang et al., 2021; T et al., 2023). However, these heavy architectures often lead to high rendering costs.

To address the trade-off between high-frequency detail transfer and real-time performance, recent work has explicitly integrated the IBR paradigm with 3D Gaussian Splatting through geometric warping. Specifically, IBGS introduces a hybrid pipeline that adopts a warping based IBR approach. It constructs warped images by projecting the intersection points of sparse Gaussians and camera rays back into source views. Instead of regressing the final colors with heavy networks, it predicts a lightweight additive residual using a CNN. While this warping based Gaussian splatting achieves an attractive quality-speed trade-off, its performance remains highly sensitive to the reliability and sharpness of the warped evidences, which can be severely degraded by geometric inaccuracies, occlusions, and mis-registrations.

## 3. Preliminaries

### 3.1. 3D Gaussian Splatting

3D Gaussian Splatting (3DGS) represents a scene as a set of 3D Gaussian primitives. Each 3D Gaussian $\mathcal{G}_i$ is parameterized by a 3D position $\boldsymbol{\mu}_i \in \mathbb{R}^3$, an opacity $o_i \in [0, 1]$, spherical harmonics (SH) coefficients $\mathbf{h}_i$, and a covariance matrix $\boldsymbol{\Sigma}_i \in \mathbb{R}^{3 \times 3}$. Following the standard formulation, the covariance is decomposed into a rotation $\mathbf{R}_i \in SO(3)$ and a diagonal scale matrix $\mathbf{S}_i \in \mathbb{R}^{3 \times 3}$ such that

$$\boldsymbol{\Sigma}_i = \mathbf{R}_i \mathbf{S}_i \mathbf{S}_i^\top \mathbf{R}_i^\top. \tag{1}$$

To render an image at a viewpoint defined by viewing transformation $\mathbf{W}$, the 3D covariance is projected to 2D covariance $\boldsymbol{\Sigma}_i^{2D}$ via EWA splatting approximation (Zwicker et al., 2002):

$$\boldsymbol{\Sigma}_i^{2D} = \mathbf{J} \mathbf{W} \boldsymbol{\Sigma}_i \mathbf{W}^\top \mathbf{J}^\top, \tag{2}$$

where $\mathbf{J}$ is the Jacobian of the affine projection approximation. Each 3D Gaussian is then splatted onto the image plane, yielding a 2D Gaussian footprint $\mathcal{G}_i^{2D}(\mathbf{p})$ at pixel $\mathbf{p}$:

$$\mathcal{G}_i^{2D}(\mathbf{p}) = \exp\left(-\frac{1}{2}(\mathbf{p} - \boldsymbol{\mu}_i^{2D})^\top (\boldsymbol{\Sigma}_i^{2D})^{-1}(\mathbf{p} - \boldsymbol{\mu}_i^{2D})\right), \tag{3}$$

where $\boldsymbol{\mu}_i^{2D}$ denotes the projected 2D mean. The base color of a pixel $\mathbf{p}$ ($\mathbf{C}_{\text{base}}(\mathbf{p})$) is then computed by front-to-back alpha compositing:

$$\mathbf{C}_{\text{base}}(\mathbf{p}) = \sum_{i \in \mathcal{N}} w_i(\mathbf{p}) \, \Psi_\ell(\mathbf{h}_i, \mathbf{v}_i(\mathbf{p})), \tag{4}$$

where $\mathcal{N}$ denotes the ordered Gaussians contributing to $\mathbf{p}$, $\mathbf{v}_i(\mathbf{p})$ is the viewing direction, and $\Psi_\ell$ maps SH coefficients $\mathbf{h}$ to an RGB color. The compositing weight is defined as

$$w_i(\mathbf{p}) = \alpha_i(\mathbf{p}) \, T_i(\mathbf{p}), \qquad T_i(\mathbf{p}) = \prod_{j=1}^{i-1} (1 - \alpha_j(\mathbf{p})), \tag{5}$$

where the effective opacity is given by $\alpha_i(\mathbf{p}) = o_i \, \mathcal{G}_i^{2D}(\mathbf{p})$. During training, all attributes including $\mathbf{h}_i$, $o_i$, $\boldsymbol{\mu}_i$, $\mathbf{q}_i$, and $\mathbf{s}_i$ are optimized end-to-end.

### 3.2. Warping based Gaussian Splatting

Warping based Gaussian Splatting via IBGS augments the standard 3DGS rendering $\mathbf{C}_{\text{base}}$ with a learnable image-space residual $\mathbf{R}_\theta$ derived from source views:

$$\mathbf{C}(\mathbf{p}) = \mathbf{C}_{\text{base}}(\mathbf{p}) + \mathbf{R}_\theta(\mathbf{p}), \tag{6}$$

To construct $\mathbf{R}_\theta$, IBGS first approximates the surface geometry. For a target pixel $\mathbf{p}$, we define the viewing ray as $\mathbf{r}(\tau) = \mathbf{o}_t + \tau \mathbf{d}_t(\mathbf{p})$, where $\mathbf{o}_t$ is the optical center, $\mathbf{d}_t(\mathbf{p})$ is the normalized ray direction, and $\tau$ denotes the distance

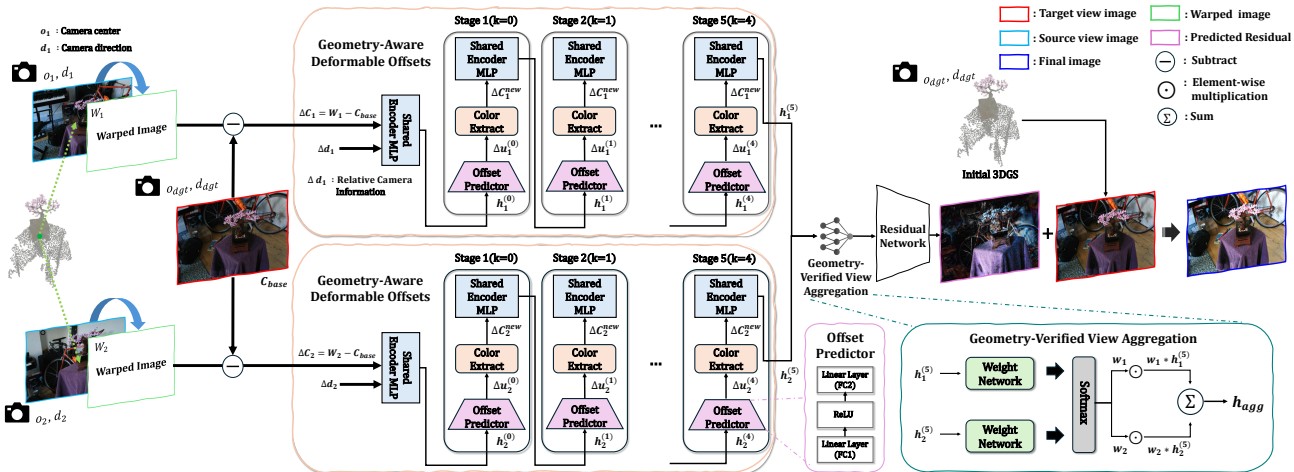

*Figure 4.* Architecture of the proposed Geometry-Aware Deformable Aggregation. To overcome the inaccuracies of geometry from given gaussian splatting, we introduce a recurrent feedback loop with shared weights. In each stage $k$, the network predicts spatial offsets ($\Delta \mathbf{u}_k$) and actively resamples the warped images to update the photometric residual ($\Delta \mathbf{C}^{new}$). This dynamic correction allows the model to look around and align features precisely before aggregation. The final refined features are combined through an adaptive attention mechanism to robustly handle occlusions and outliers.

along the ray. Proxy surface points $\mathbf{x}_i$ are obtained by intersecting this ray with the planes of a selected subset of Gaussians $i \in \mathcal{S}(\mathbf{p})$ (typically near the transmittance median), defined by their centers $\boldsymbol{\mu}_i$ and learnable normals $\mathbf{n}_i$:

$$\mathbf{x}_i(\mathbf{p}) = \mathbf{o}_t + \frac{\mathbf{n}_i^\top (\boldsymbol{\mu}_i - \mathbf{o}_t)}{\mathbf{n}_i^\top \mathbf{d}_t(\mathbf{p})} \mathbf{d}_t(\mathbf{p}), \tag{7}$$

These points are projected into source view $m$ via $\pi_m$ to sample colors $\tilde{\mathbf{c}}_{i,m}$ from the source image $\mathbf{I}_m$ using a differentiable sampler $\mathcal{B}$ (i.e., $\tilde{\mathbf{c}}_{i,m} = \mathcal{B}(\mathbf{I}_m, \pi_m(\mathbf{x}_i))$). The warped image $\mathbf{W}_m$ is then composed by aggregating these samples utilizing the standard 3DGS blending weights $w_i(\mathbf{p})$:

$$\mathbf{W}_m(\mathbf{p}) = \frac{\sum_{i \in \mathcal{S}(\mathbf{p})} w_i(\mathbf{p}) \tilde{\mathbf{c}}_{i,m}(\mathbf{p})}{\sum_{i \in \mathcal{S}(\mathbf{p})} w_i(\mathbf{p}) + \epsilon}. \tag{8}$$

Finally, the residual is predicted by processing per-view features based on the appearance difference $\Delta \mathbf{c}_m = \mathbf{W}_m - \mathbf{C}_{\text{base}}$ and relative pose $\Delta \mathbf{d}_m$ through a lightweight MLP, followed by pooling and a CNN decoder

## 4. Method

Given unstructured source images $\mathcal{I} = \{I_1, \ldots, I_N\}$ and uncertain geometry $\mathcal{G}$, we propose a warping-based Gaussian Splatting framework to synthesize high-fidelity novel views. To incorporate fine-grained details from source views, we render a base color $\mathbf{C}_{\text{base}}$ and project surface intersections $\mathbf{x} \in \mathcal{G}$ onto source views, yielding sampling coordinates $\mathbf{u}_m = \pi_m(\mathbf{x})$.

However, uncertainty in the geometry $\mathcal{G}$ inevitably misaligns $\mathbf{u}_m$ from the true correspondence. Instead of simply discarding these pixels as outliers, which leads to information

loss, we reinterpret the error as coherent spatial misalignments, positing that valid visual cues are locally preserved near $\mathbf{u}_m$. GADA recovers these cues through a progressive three-stage pipeline: First, Geometric Context Embedding (Sec. 4.1) constructs a robust state vector by combining photometric error with geometric context to guide the search. Next, Geometry-Aware Deformable Offsets (Sec. 4.2) iteratively predicts offsets $\Delta \mathbf{u}$ to actively retrieve aligned textures from the local neighborhood. Finally, Geometry-Verified View Aggregation (Sec. 4.3) resolves remaining ambiguities via adaptive weighting, effectively preventing detail dilution and ensuring high-fidelity results.

### 4.1. Geometric Context Embedding

To learn the deformation offsets effectively, the network requires a robust state representation that captures both the current alignment error and the local geometric reliability. Formally, given an uncertain geometry-projected coordinate $\mathbf{u}_m^{(0)} = \pi_m(\mathbf{x})$, we construct an input vector containing the appearance residual $\Delta \mathbf{c}_m$ and the geometric context $\Delta \mathbf{d}_m$. This input is processed by a shared encoder MLP $\Phi_{\text{enc}}$ to produce the initial latent state $\mathbf{h}_m^{(0)}$: $\mathbf{h}_m^{(0)} = \Phi_{\text{enc}}\left( \left[ \Delta \mathbf{c}_m(\mathbf{u}_m^{(0)}) \oplus \Delta \mathbf{d}_m \right] \right)$. Here, $\oplus$ denotes concatenation. The first component, Appearance Residual $\Delta \mathbf{c}_m = \mathbf{W}_m(\mathbf{u}_m) - \mathbf{C}_{\text{base}}$, explicitly provides the magnitude of the initial photometric misalignment.

The second component, Geometric Context $\Delta \mathbf{d}_m$, encodes the spatial relationship between the source and target views. Consistent with the preliminary definition, we instantiate $\Delta \mathbf{d}_m$ as a 4-dimensional vector combining relative translation and viewing angle difference: $\Delta \mathbf{d}_m = [(\mathbf{o}_m - \mathbf{o}_{\text{tgt}}) \oplus (\mathbf{d}_m \cdot \mathbf{d}_{\text{tgt}})]$. Here, $\mathbf{o}_m$ and $\mathbf{o}_{\text{tgt}}$ denote the optical centers

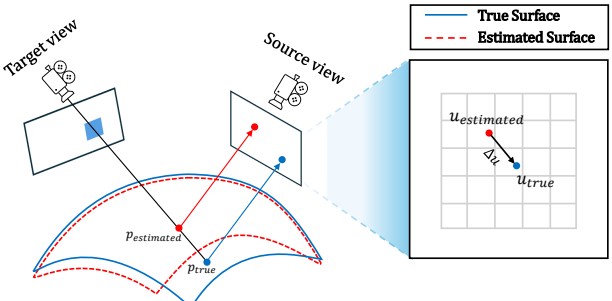

*Figure 5.* Geometry induced spatial misalignments in warped evidences.

of the source and target views, respectively, while $\mathbf{d}_m$ and $\mathbf{d}_{\text{tgt}}$ represent their normalized principal viewing directions. By encoding these relative pose cues alongside color errors, $\mathbf{h}_m^{(0)}$ serves as a robust starting point for our iterative correction process.

### 4.2. Geometry-Aware Deformable Offset

As shown in Fig. 5, geometry errors displace the projected sampling location from the true correspondence. This deviation manifests as a coherent local misalignment rather than random noise. To rectify this, GADA introduces an iterative alignment module. Unlike standard methods that rely on fixed coordinates, our approach predicts bounded offsets conditioned on geometric context. This active search retrieves valid evidence from the local neighborhood, enabling cleaner residual learning of high-frequency details.

**Geometry-Aware Offset Prediction.** Let $\mathbf{u}_m^{(k)}$ denote the sampling coordinate at iteration $k$, initialized as $\mathbf{u}_m^{(0)} = \pi_m(\mathbf{x})$. At step $k$, given the current latent state $\mathbf{h}_m^{(k)}$, the lightweight offset network $\mathcal{F}_{\text{off}}$ predicts a displacement field :

$$\Delta\mathbf{u}_m^{(k)}(\mathbf{p}) = \sigma_{\max} \cdot \tanh\left(\mathcal{F}_{\text{off}}(\mathbf{h}_m^{(k)})\right), \qquad (9)$$

We then update the sampling coordinate: $\mathbf{u}_m^{(k+1)} \leftarrow \mathbf{u}_m^{(k)} + \Delta\mathbf{u}_m^{(k)}$. The $\tanh$ activation with scaling $\sigma_{\max}$ ensures the search adheres to the local neighborhood.

**Color Extract and Recurrent Update.** Using the updated coordinate $\mathbf{u}_m^{(k+1)}$, we actively resample the source image to extract the corrected color $\tilde{\mathbf{c}}_m^{(k+1)}$. This new observation serves as feedback for the next iteration. We update the appearance residual: $\Delta\mathbf{c}_m^{(k+1)} = \tilde{\mathbf{c}}_m^{(k+1)} - \mathbf{C}_{\text{base}}$. This residual represents the remaining error *after* the correction. To close the loop, we feed this new residual back into the same shared encoder $\Phi_{\text{enc}}$ alongside the fixed geometric context $\Delta\mathbf{d}_m$: $\mathbf{h}_m^{(k+1)} = \Phi_{\text{enc}}\left(\left[\Delta\mathbf{c}_m^{(k+1)} \oplus \Delta\mathbf{d}_m\right]\right)$. By repeating this process for $k = 0, \dots, K-1$, the network progressively minimizes the photometric error. The final feature $\mathbf{h}_m^{(K)}$ thus encodes both the refined local texture and the confidence of the geometric alignment.

### 4.3. Geometry-Verified View Aggregation

After $K$ iterations, we obtain the refined features from multiple source views. We must then aggregate them to synthesize the final target ray color. Even with optimal deformation, ambiguities such as occlusions remain. To handle this, we propose an uncertainty-aware aggregation mechanism.

**Confidence Estimation.** We estimate a confidence score for each view based on the final refined feature $\mathbf{h}_m^{(K)}$. Since $\mathbf{h}_m^{(K)}$ encodes the post-correction residual, it serves as a strong indicator of geometric validity. A weight network $\mathcal{F}_{\text{w}}$ predicts the unnormalized log-probability, which is normalized via Softmax:

$$w_m = \text{Softmax}_m(\mathcal{F}_{\text{w}}(\mathbf{h}_m^{(K)})), \qquad (10)$$

This mechanism effectively down-weights views that fail to align photometrically (large residual in $\mathbf{h}_m^{(K)}$) or are geometrically invalid (indicated by $\Delta\mathbf{d}_m$ within $\mathbf{h}_m^{(K)}$).

**Late Fusion and Decoding.** Finally, we compute the aggregated consensus feature $\mathbf{h}_{\text{agg}}$ via a weighted sum:

$$\mathbf{h}_{\text{agg}} = \sum_{m=1}^{M} w_m \cdot \mathbf{h}_m^{(K)}. \qquad (11)$$

To synthesize the final residual color $\Delta\mathbf{C}(\mathbf{r})$, we employ a Residual CNN decoder $\mathcal{D}$. The aggregated feature is conditioned on the target ray direction $\mathbf{d}$ and the base Gaussian color $\mathbf{C}_{\text{base}}$ to recover high-frequency details: $\Delta\mathbf{C}(\mathbf{r}) = \mathcal{D}(\mathbf{h}_{\text{agg}})$. This late-fusion architecture ensures that the final synthesis is robust to individual view outliers while preserving the global structural coherence.

### 4.4. Optimization Objectives

Our network is optimized end-to-end to enforce both photometric fidelity and geometric consistency. Following the baseline strategy (Nguyen et al., 2026), the total objective $\mathcal{L}$ is defined as a weighted sum of reconstruction losses and geometric regularizers:

$$\mathcal{L} = \mathcal{L}_{\text{rgb}} + \lambda_1 \mathcal{L}_{\text{photo}} + \lambda_2 \mathcal{E}_{\text{reg}} + \lambda_3 \mathcal{L}_{\text{normal}}, \qquad (12)$$

**Reconstruction and Consistency Losses.** We adopt the standard loss terms from IBGS and 2DGS (Huang et al., 2024). Specifically, $\mathcal{L}_{\text{rgb}}$ minimizes the $\mathcal{L}_1$ and D-SSIM errors for both the base Gaussian rendering and the final rectified image. The multi-view consistency loss $\mathcal{L}_{\text{photo}}$ ensures that the warped source patches align with the target view, supervising the deformation field to sample physically meaningful locations. Additionally, we incorporate the normal consistency loss $\mathcal{L}_{\text{normal}}$ from 2DGS to align rendered normals with pseudo-normals derived from depth gradients, reducing high-frequency geometric artifacts.

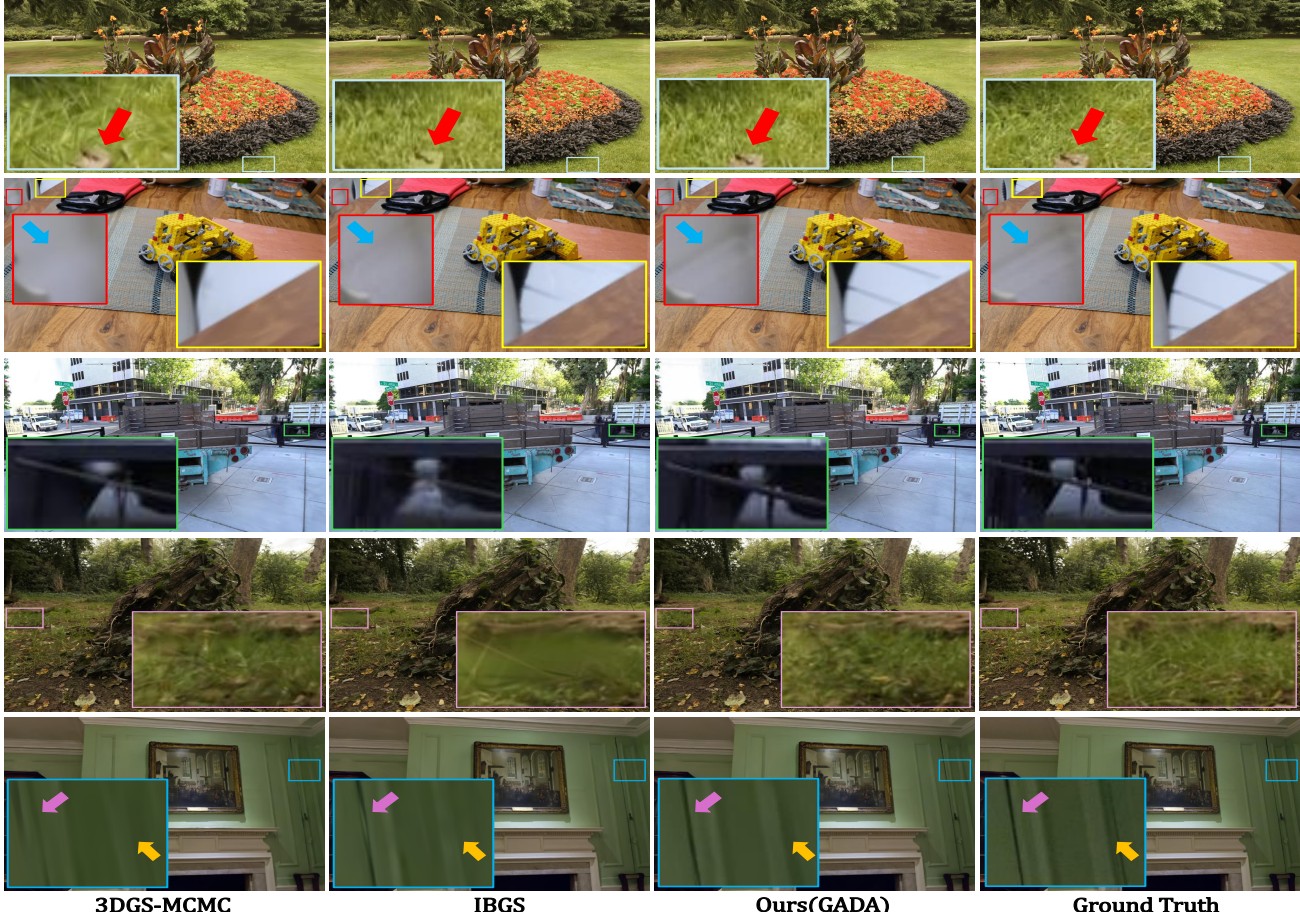

|      3DGS-MCMC      |      IBGS      |      Ours(GADA)      |      Ground Truth      |

*Figure 6.* Qualitative comparison of novel view synthesis on benchmark datasets. From left to right: 3DGS-MCMC, IBGS, Ours, and Ground Truth. As highlighted in the zoomed-in patches, our method reconstructs fine geometric details and textures more accurately than the baselines, closely matching the ground truth.

**Geometric Elastic Regularization ($\mathcal{E}_{\mathbf{reg}}$).** While photometric terms drive the alignment, unconstrained deformation can lead to geometric instabilities or topological tearing in textureless regions. To mitigate this, we introduce an elastic regularization penalty on the displacement vectors:

$$\mathcal{E}_{\text{reg}} = \frac{1}{N} \sum_m \|\Delta \mathbf{u}_m\|_2^2. \tag{13}$$

where $N$ denotes the total number of sampled pixels. This term acts as a geometric anchor, encouraging the deformation field to adhere to the global structure provided by the proxy geometry unless there is a strong photometric gradient to justify a shift.

## 5. Experiment

### 5.1. Experimental Settings

**Implementation Details.** We follow the standard hyperparameter configurations of the baseline methods. Specifically, we set the number of refinement iterations to $K = 5$, the geometric regularization weight to $\lambda_2 = 0.01$, and the maximum offset magnitude to $\sigma_{\max} = 7$ for bounded local refinement (see Appendix for sensitivity). Our model is trained end-to-end for 30,000 iterations using the Adam optimizer (Kingma, 2014). To ensure geometric stability, we adopt a warm-up phase for the deformation module in the early training stages (first 3,000 iterations). Standard data augmentation and densification strategies are applied following prior works. All experiments are conducted on a single NVIDIA A100 GPU.

**Dataset** We evaluate our method on standard NVS benchmarks: Mip-NeRF 360 (Barron et al., 2022) (9 scenes), Tanks and Temples (Knapitsch et al., 2017) (2 scenes), and Deep Blending (Hedman et al., 2018) (2 scenes). Additionally, to validate performance on challenging view-dependent effects such as specular highlights, we include 3 scenes from the Shiny dataset (Wizadwongsa et al., 2021). Following standard protocols, we hold out every $8^{th}$ image for evaluation and use the remaining images for training. For comparison, we benchmark GADA against recent state-of-the-art methods, demonstrating the effectiveness of our geometry-aware aggregation.

*Table 1.* Quantitative comparison with state-of-the-art methods on three benchmark datasets. We report PSNR, SSIM, and LPIPS metrics. **Bold** indicates the best performance. We also highlight the best , second , and third results with background colors.

| Method | Mip-NeRF 360 | | | Tanks & Temples | | | Deep Blending | | |
|---|---|---|---|---|---|---|---|---|---|
| | PSNR↑ | SSIM↑ | LPIPS↓ | PSNR↑ | SSIM↑ | LPIPS↓ | PSNR↑ | SSIM↑ | LPIPS↓ |
| 2DGS (Huang et al., 2024) | 26.81 | 0.796 | 0.297 | 23.13 | 0.833 | 0.211 | 29.49 | 0.903 | 0.256 |
| PGSR (Chen et al., 2024) | 27.20 | 0.819 | 0.233 | 24.20 | 0.857 | 0.167 | 29.22 | 0.894 | 0.255 |
| Taming 3DGS (Mallick et al., 2024) | 27.21 | 0.793 | 0.305 | 24.04 | 0.851 | 0.170 | 29.87 | 0.907 | 0.235 |
| Octree-GS (Ren et al., 2025) | 27.39 | 0.811 | 0.264 | 24.52 | 0.866 | 0.153 | 30.41 | 0.913 | 0.238 |
| 3DGS (Kerbl et al., 2023) | 27.43 | 0.814 | 0.257 | 23.14 | 0.840 | 0.183 | 29.4 | 0.902 | 0.242 |
| Mip-Splatting (Yu et al., 2024) | 27.49 | 0.815 | 0.258 | 23.74 | 0.859 | 0.156 | 29.35 | 0.902 | 0.238 |
| Scaffold-GS (Lu et al., 2024) | 27.71 | 0.813 | 0.262 | 23.96 | 0.852 | 0.177 | 30.21 | 0.906 | 0.254 |
| AbsGS (Ye et al., 2024) | 27.49 | 0.820 | 0.191 | 23.73 | 0.853 | 0.162 | 29.67 | 0.902 | 0.236 |
| 3DGS-MCMC (Kheradmand et al., 2024) | 27.98 | 0.835 | 0.224 | 24.29 | 0.860 | 0.190 | 29.67 | 0.895 | 0.320 |
| IBGS (Nguyen et al., 2026) | 28.29 | 0.831 | 0.191 | 24.75 | 0.861 | 0.154 | 29.94 | 0.899 | 0.237 |
| **Ours** | **28.62** | **0.840** | **0.179** | **24.92** | **0.871** | **0.144** | 30.22 | 0.911 | 0.235 |

## 5.2. Evaluation Metrics

We evaluate the rendering performance based on three primary qualities: (1) photometric accuracy, (2) perceptual fidelity, and (3) computational efficiency. The photometric accuracy measures the pixel-level signal fidelity between the rendered images and ground truth using Peak Signal-to-Noise Ratio (PSNR) and Structural Similarity Index Measure (SSIM) (Wang et al., 2004). The perceptual fidelity assesses the recovery of high-frequency details and texture realism, utilizing the Learned Perceptual Image Patch Similarity (LPIPS) metric (Zhang et al., 2018). Finally, the computational efficiency evaluates the practicality of the method by measuring rendering speed (Frames Per Second, FPS), training time, and storage consumption compared to baseline methods.

## 5.3. Experimental Results

**Qualitative Comparisons.** Figure 6 compares GADA against state-of-the-art methods on challenging scenes. As shown in the second column, IBGS suffers from blurring and ghosting artifacts (e.g., discolored leaves) due to its reliance on uncertain geometry, which prevents the aggregation of valid source features. In contrast, GADA actively rectifies these local misalignments via learnable offsets, successfully recovering high-frequency details by retrieving otherwise discarded information for robust residual learning. Similarly, while 3DGS-MCMC maintains global consistency, it tends to over-smooth thin structures, causing the metal bars in the $3^{rd}$ row to appear disconnected. Our method effectively mitigates this by adapting the receptive field to geometric errors, delivering sharp structural connectivity. Furthermore, GADA faithfully preserves complex textures (e.g., the tree bark in the $4^{th}$ row). This improvement stems from our geometry-aware mechanism, which actively searches for valid cues unlike previous methods, ensuring consistent detail restoration even with uncertain geometry.

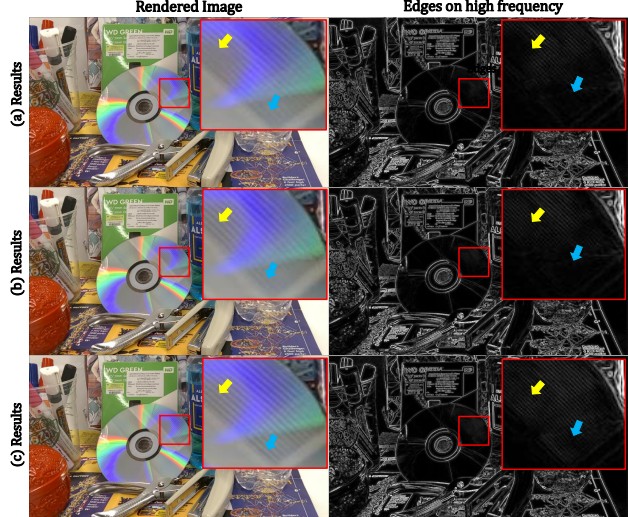

*Figure 7.* **Visual ablation of component contributions.** (a) The Baseline suffers from blur due to geometric misalignment. (b) Adding Geometry-Verified View Aggregation improves signal consistency. (c) The Full Model, equipped with offset prediction, successfully recovers high-frequency details and sharp edges.

*Table 2.* Ablation study on the number of iterations $K$ evaluated on Mip-NeRF 360. We select $K = 5$ as the optimal configuration.

| Iterations $(K)$ | PSNR↑ | SSIM↑ | LPIPS↓ | FPS↑ |
|---|---|---|---|---|
| 0 | 28.33 | 0.834 | 0.190 | **61** |
| 1 | 28.42 | 0.835 | 0.186 | 58 |
| 3 | 28.55 | 0.838 | 0.182 | 52 |
| **5** | **28.62** | **0.840** | **0.179** | 47 |
| 7 | 28.62 | 0.841 | 0.179 | 42 |

**Quantitative Results.** We evaluate GADA against state-of-the-art methods on benchmarks: Mip-NeRF 360, Tanks & Temples, and Deep Blending (Table 1). Our method achieves state-of-the-art or competitive performance. Although Octree-GS shows marginally higher PSNR on Deep Blending, our method attains the lowest LPIPS, demonstrat-

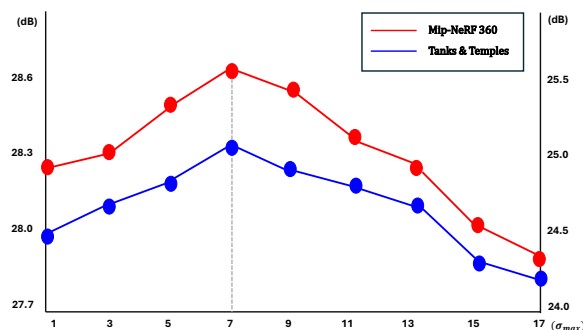

Figure 8. **Impact of $\sigma_{\max}$ on reconstruction quality.** A value of 7 yields the highest PSNR

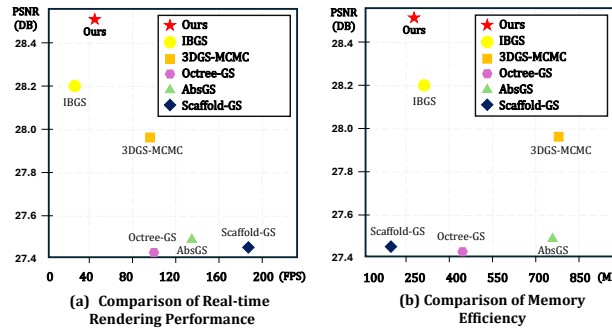

Figure 9. **Efficiency Comparison on Mip-NeRF 360.** (a) and (b) show quantitative comparisons of rendering quality and efficiency from 47 to 42. Therefore, we adopt $K = 5$ as the optimal configuration, balancing high-fidelity reconstruction with real-time performance.

ing superior perceptual fidelity. This indicates that image-space residuals effectively recover high-frequency details challenging for primitives. Furthermore, a direct comparison with IBGS demonstrates the efficacy of our geometry-aware aggregation; while IBGS suffers from limiting perceptual quality due to spatial misalignments, GADA effectively rectifies these errors, substantially improving LPIPS, confirming our learnable offsets and confidence weighting recover valid cues lost by geometric inaccuracies.

### 5.4. Ablation Studies

We conduct an analysis to validate the effectiveness of our proposed components, focusing on reconstruction quality, hyperparameter sensitivity[4], and computational efficiency.

**Impact of Geometry-Aware Components.** First, we visually examine the contribution of each module in Figure 7. The baseline model (Figure 7a), which relies solely on fixed geometric proxies, suffers from significant misalignment. Consequently, high-frequency reflections are heavily blurred, making the ventilation fan reflected on the CD surface indistinguishable. Adding the Geometry-Verified View Aggregation (Figure 7b) improves signal consistency by suppressing outlier colors, yet it fails to resolve fine structural details. In contrast, the full model equipped with learnable offsets (Figure 7c) drastically improves reconstruction. By actively deforming sampling coordinates to align with the true surface, our method successfully recovers intricate details, clearly revealing the structure of the ventilation fan and the sharp edges of the pliers.

**Number of Deformation Iterations ($K$).** To investigate the efficacy of our iterative refinement strategy, we evaluate performance across varying numbers of deformation steps $K$, as summarized in Table 2. Increasing $K$ allows the network to progressively refine sampling coordinates, leading to consistent improvements in PSNR and SSIM. However, this comes at the cost of rendering speed. We observe that the quality gain saturates at $K = 5$, where increasing to $K = 7$ yields no further PSNR improvement while dropping FPS

**Sensitivity of $\sigma_{\max}$.** We investigate the optimal search range by varying $\sigma_{\max}$ across benchmark datasets. As shown in Fig. 8, performance peaks at $\sigma_{\max} = 7$, representing an optimal trade-off. A search radius smaller than 7 is insufficient to correct significant geometric misalignments, whereas a larger radius ($\sigma_{\max} > 7$) degrades quality by retrieving irrelevant textures from distant regions. Consequently, we adopt $\sigma_{\max} = 7$ as the default setting.

**Efficiency Trade-off Analysis.** Finally, Figure 9 demonstrates that GADA achieves a superior efficiency trade-off on the Mip-NeRF 360 dataset compared to state-of-the-art methods.[5] As shown in Figure 9(a), our method (red star) occupies the optimal upper-right region. Notably, we achieve a rendering framerate of 47 FPS, which is more than $2\times$ faster than the baseline IBGS (22 FPS), validating the computational efficiency of our lightweight recurrent architecture. Furthermore, Figure 9(b) highlights our spatial efficiency. Our method resides in the high-fidelity, low-storage region, requiring significantly less memory than standard 3DGS variants and remaining comparable to the highly compact Scaffold-GS. This confirms that GADA enables high-fidelity rendering without the excessive storage costs typically associated with primitive densification.

**Matched Training Budget.** While the previous analysis focuses on rendering FPS and storage efficiency, we further examine whether the additional recurrent deformable refinement introduces a prohibitive training or memory cost. This analysis is important because GADA performs iterative offset refinement during optimization, which could appear to increase computational complexity compared to standard 3DGS or IBGS. We therefore evaluate the trade-off from two complementary perspectives: matched training budget and peak GPU memory usage. GADA requires a longer training time than vanilla 3DGS and IBGS because the deformation module is optimized through recurrent refinement. However, this additional cost directly contributes

---

[4]Additional sensitivity analysis on $\lambda_2$ is provided in Appendix B.2.

[5]Detailed results are provided in the Appendix D.3.

*Table 3.* **Comparison under matched training budgets.** Extended-budget baselines improve only marginally with longer optimization, whereas GADA consistently achieves better reconstruction quality under comparable wall-clock time.

| Method | Mip-NeRF 360 | | | | Tanks&Temples | | | | Deep Blending | | | |
|---|---|---|---|---|---|---|---|---|---|---|---|---|
| | PSNR↑ | SSIM↑ | LPIPS↓ | Time↓ | PSNR↑ | SSIM↑ | LPIPS↓ | Time↓ | PSNR↑ | SSIM↑ | LPIPS↓ | Time↓ |
| 3DGS | 27.43 | 0.814 | 0.257 | 30m | 23.14 | 0.840 | 0.183 | 13m | 29.40 | 0.902 | 0.242 | 36m |
| 3DGS-Long | 27.57 | 0.817 | 0.252 | 55m | 23.36 | 0.846 | 0.179 | 35m | 29.50 | 0.903 | 0.240 | 42m |
| IBGS | 28.29 | 0.831 | 0.191 | 43m | 24.75 | 0.861 | 0.154 | 24m | 29.94 | 0.899 | 0.237 | 37m |
| IBGS-Long | 28.33 | 0.834 | 0.188 | 52m | 24.79 | 0.863 | 0.152 | 33m | 29.95 | 0.901 | 0.235 | 41m |
| **Ours** | **28.62** | **0.840** | **0.179** | 52m | **24.92** | **0.871** | **0.144** | 30m | **30.22** | **0.911** | **0.235** | 42m |

*Table 4.* **Peak VRAM comparison.** GADA reduces peak memory usage by removing visibility-check-related buffers.

| Dataset | Inference GB↓ | | Training GB↓ | |
|---|---|---|---|---|
| | IBGS | Ours | IBGS | Ours |
| Mip-NeRF 360 | 6.12 | **4.47** | 6.48 | **5.91** |
| Tanks&Temples | 2.97 | **2.27** | 3.46 | **3.10** |
| Deep Blending | 5.36 | **4.08** | 5.94 | **5.36** |

to more accurate local cue retrieval and improved residual prediction. To verify that the improvement is not simply caused by longer optimization, we compare GADA with extended-budget variants of 3DGS and IBGS under similar wall-clock training time. As shown in Table 3, even when 3DGS and IBGS are trained for longer, GADA consistently achieves better reconstruction quality across Mip-NeRF 360, Tanks&Temples, and Deep Blending.

The results show that simply extending the optimization time of existing methods does not close the gap. For example, on Mip-NeRF 360, extending IBGS from 43 minutes to 52 minutes improves PSNR only from 28.29 to 28.33, whereas GADA achieves 28.62 under the same 52-minute budget. A similar trend is observed on Tanks&Temples and Deep Blending. This suggests that the performance gain of GADA comes from the proposed geometry-aware deformable aggregation itself, rather than from additional training iterations alone.

**Peak GPU Memory Usage.** We also analyze peak VRAM usage to examine whether the recurrent module increases the memory burden. Although GADA introduces additional deformable-offset computation, it removes the explicit visibility-check components required by IBGS, such as depth-list buffers and source re-rendering buffers. As summarized in Table 4, this design leads to lower peak memory usage in both inference and training.

These results indicate that the recurrent deformable refinement does not translate into higher overall memory consumption. In inference, GADA avoids the depth-list and source re-rendering buffers used by IBGS, requiring only a lightweight deformable-offset module. During training,

the additional backpropagation cost of the deformable module is also smaller than the memory saved by removing the visibility-check buffers. Therefore, GADA provides a favorable practical trade-off: it introduces a moderate training-time overhead, but achieves higher rendering quality, over $2\times$ faster inference than IBGS, lower peak VRAM usage, and compact storage requirements.

### 5.5. Future Work

Future work could extend GADA in three directions. First, while our local deformable refinement is effective under standard novel-view synthesis settings, extremely sparse-view scenarios may require broader correspondence search or global matching priors to handle severe disocclusions and large angular gaps. Second, the refinement process could be made adaptive by varying $K$ and $\sigma_{\max}$ across pixels or scenes, allocating computation only to regions with high geometric uncertainty. Third, lightweight offset predictors, early-exit refinement, or distillation-based acceleration could further reduce training cost while preserving high-frequency detail recovery. Extending GADA to cross-scene generalization, dynamic scenes, and video-consistent rendering would also broaden its practical applicability.

## 6. Conclusion

We presented GADA to address the trade-off between rendering quality and efficiency in warping-based Gaussian Splatting. By identifying geometric misalignment as a key bottleneck, GADA replaces heuristic visibility checks with geometry-aware deformable offsets that actively retrieve locally displaced visual cues. Together with geometry-verified view aggregation, this enables robust residual prediction by preserving reliable high-frequency evidence while suppressing unreliable warped inputs. Experiments on standard novel-view synthesis benchmarks show that GADA achieves state-of-the-art rendering quality with practical efficiency, reaching 47 FPS and providing over a $2\times$ speedup over IBGS while maintaining a compact memory footprint. Overall, our results demonstrate that lightweight residual correction guided by local geometric refinement can enhance fidelity without excessive storage or inference cost.

## Acknowledgement

This work was supported by Institute for Information & communications Technology Planning & Evaluation (IITP) grant funded by the Korea government (MSIT) (No. RS-2021-II211381, Development of Causal AI through Video Understanding and Reinforcement Learning, and Its Applications to Real Environments) and the Institute of Information & communications Technology Planning & Evaluation (IITP) grant funded by the Korea government (MSIT) (No. RS-2022-II220184, Development and Study of AI Technologies to Inexpensively Conform to Evolving Policy on Ethics).

## Impact Statement

This research presents GADA, a novel framework that significantly advances the fidelity and efficiency of 3D scene reconstruction. By overcoming the fundamental limitations of warping-based methods specifically the loss of high-frequency details and computational bottlenecks our approach enables the synthesis of highly detailed 3D environments in real-time. This advancement has a direct impact on various fields, including Virtual and Augmented Reality (VR/AR), where photorealistic immersion is critical, and Digital Twin technology, which requires precise geometric modeling. Furthermore, by eliminating heuristic-heavy processes, GADA offers a more robust and scalable solution for large-scale 3D content creation, setting a new benchmark for high-quality, efficient view synthesis in the evolving landscape of computer vision.

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

# A. Analysis of Explicit Visibility Checks vs GADA

In this section, we analyze the limitations of explicit visibility checks employed in baseline methods (e.g., IBGS (Nguyen et al., 2026)) and demonstrate why our geometry-aware deformable aggregation offers a numerically superior alternative.

## A.1. Formulation of Explicit Visibility Check in IBGS

Baseline approaches typically rely on a depth-consistency test to filter out occluded source views. Specifically, IBGS employs a relative depth difference check. Let $z(\mathbf{x}(p))$ be the depth of the proxy surface point $\mathbf{x}$ projected onto the target view. To determine if a source view $s$ is visible, they compare this proxy depth with the depth value $z(\mathbf{x}_s^{\text{warp}}(p))$ retrieved from the source view's depth map:

$$\frac{|z(\mathbf{x}(p)) - z(\mathbf{x}_s^{\text{warp}}(p))|}{z(\mathbf{x}(p)) + z(\mathbf{x}_s^{\text{warp}}(p))} \leq \tau, \tag{14}$$

where $\tau$ is a pre-defined depth error threshold (typically set to 0.001) and $\mathbf{x}_s^{\text{warp}}(p)$ is the point sampled from the source view's proxy geometry. Features from source view $s$ are strictly discarded if this condition is not met.

## A.2. The Computational Cost of Explicit Checks

While the visibility condition in Eq. 14 theoretically ensures geometric consistency, its practical implementation introduces a severe computational bottleneck during inference. To evaluate the inequality, the system requires immediate access to the source view's depth $z(\mathbf{x}_s^{\text{warp}}(p))$. According to the implementation details of IBGS, while these depth maps can be precomputed and stored after each training iteration, they are not readily available for novel view synthesis. Consequently, at test time, depth maps of source images must be rendered on-the-fly to perform the visibility check against the proxy geometry. This necessitates additional rasterization passes for every selected source view before any feature aggregation can occur. Furthermore, this process requires modifying CUDA kernels to support ray-Gaussian intersections and per-pixel visibility handling, which significantly complicates the rendering pipeline and increases memory latency.

## A.3. Efficiency of GADA via Implicit Weighting

In contrast, GADA eliminates the dependency on explicit depth comparison (Eq. 14), thereby bypassing the need for run-time depth rasterization. Instead of a hard boolean check, our **Geometric Context Embedding** and **Confidence Estimation** modules implicitly determine the reliability of source features. Since the confidence weights are predicted directly from feature discrepancies and rela-

tive poses via a lightweight MLP, our method avoids the overhead of on-the-fly rendering. Table 5 summarizes this algorithmic distinction. Our design allows GADA to maintain high-speed rendering (47 FPS) by accessing source features directly, whereas the baseline IBGS experiences a substantial slowdown (22 FPS) due to the mandatory depth verification steps.

*Table 5.* Ablation study on visibility handling strategies on Mip-NeRF 360. Removing explicit checks doubles the rendering speed while improving quality.

| Method | PSNR↑ | SSIM↑ | LPIPS↓ | FPS↑ |
|---|---|---|---|---|
| IBGS w/ Vis. Check | 28.29 | 0.831 | 0.191 | 22 |
| GADA w/ Vis. Check | 28.45 | 0.836 | 0.185 | 18 |
| **GADA w/o Vis. Check** | **28.62** | **0.840** | **0.179** | **47** |

*Table 6.* Quantitative comparison on Tanks & Temples. Our implicit weighting strategy significantly outperforms the explicit check baseline.

| Method | PSNR↑ | SSIM↑ | LPIPS↓ | FPS↑ |
|---|---|---|---|---|
| IBGS w/ Vis. Check | 24.75 | 0.861 | 0.154 | 24 |
| GADA w/ Vis. Check | 24.85 | 0.866 | 0.148 | 21 |
| **GADA w/o Vis. Check** | **24.92** | **0.871** | **0.142** | **52** |

*Table 7.* Quantitative comparison on Deep Blending. Even with comparable PSNR, removing explicit checks ensures higher inference speed.

| Method | PSNR↑ | SSIM↑ | LPIPS↓ | FPS↑ |
|---|---|---|---|---|
| IBGS w/ Vis. Check | 30.08 | 0.899 | 0.237 | 28 |
| GADA w/ Vis. Check | 30.15 | 0.905 | 0.236 | 25 |
| **GADA w/o Vis. Check** | **30.22** | **0.911** | **0.235** | **58** |

# B. Additional Experiments and Ablation

In this section, we provide further analysis of our method's design choices, specifically focusing on the number of refinement iterations and the robustness of our approach in challenging specular scenarios.

## B.1. Evaluation on Specular Scenes (Shiny Dataset)

To further demonstrate the robustness of GADA in handling view-dependent effects, specifically high-frequency specular highlights, we evaluate our method on the Shiny dataset (Wizadwongsa et al., 2021). This dataset contains scenes with complex reflections (e.g., *CD*, *Lab*) that are particularly challenging for methods relying on fixed proxy geometries.

Table 8 presents the quantitative comparison against baseline methods. Standard 3DGS often creates "foggy" artifacts around specular surfaces due to the limitation of Spherical Harmonics in modeling high-frequency reflections. IBGS

improves upon this but suffers from artifacts when the proxy surface geometry is slightly misaligned with the reflection plane. In contrast, GADA significantly outperforms baselines across all metrics. By actively searching for the correct reflection features in the source views via learnable offsets, our method accurately reconstructs sharp specular highlights and intricate reflected details, as qualitatively shown in the main paper.

*Table 8.* **Quantitative comparison on the Shiny dataset.** We compare GADA against Spec-Gauss (Yang et al., 2024) on three challenging scenes. GADA outperforms the baseline in most metrics, particularly in PSNR and LPIPS.

| Scene | Method | PSNR↑ | SSIM↑ | LPIPS↓ |
|---|---|---|---|---|
| **Guitars** | Spec-Gauss (Yang et al., 2024) | 30.62 | **0.955** | 0.120 |
| | IBGS | 35.78 | 0.954 | 0.105 |
| | **GADA (Ours)** | **35.93** | 0.953 | **0.104** |
| **Lab** | Spec-Gauss (Yang et al., 2024) | 30.53 | 0.946 | 0.103 |
| | IBGS | 35.06 | **0.966** | **0.056** |
| | **GADA (Ours)** | **35.32** | 0.965 | **0.056** |
| **CD** | Spec-Gauss (Yang et al., 2024) | 30.69 | 0.954 | 0.081 |
| | IBGS | 35.23 | 0.955 | 0.060 |
| | **GADA (Ours)** | **35.45** | **0.956** | **0.056** |

## B.2. Effectiveness of loss function

*Table 9.* **Ablation on regularization weight** $\lambda_2$ **(Mip-NeRF 360).** Setting $\lambda_2 = 0.01$ provides the optimal balance between geometric freedom and stability.

| $\lambda_2$ | PSNR↑ | SSIM↑ | LPIPS↓ |
|---|---|---|---|
| 0 (w/o) | 28.15 | 0.825 | 0.195 |
| 0.001 | 28.50 | 0.835 | 0.183 |
| **0.01 (Default)** | **28.62** | **0.840** | **0.179** |
| 0.1 | 28.41 | 0.833 | 0.188 |
| 1.0 | 28.20 | 0.820 | 0.210 |

*Table 10.* **Ablation on regularization weight** $\lambda_2$ **(Tanks & Temples).** A moderate penalty ($\lambda_2 = 0.01$) effectively suppresses noise without over-constraining the search.

| $\lambda_2$ | PSNR↑ | SSIM↑ | LPIPS↓ |
|---|---|---|---|
| 0 (w/o) | 24.40 | 0.855 | 0.160 |
| 0.001 | 24.81 | 0.866 | 0.149 |
| **0.01 (Default)** | **24.92** | **0.871** | **0.144** |
| 0.1 | 24.75 | 0.865 | 0.152 |
| 1.0 | 24.30 | 0.850 | 0.175 |

We analyze the sensitivity of our model to the geometric elastic regularization weight $\lambda_2$, which governs the flexibility of the deformation offset. As detailed in Tables 9, 10, and 11, we observe a consistent trade-off across all datasets. When regularization is negligible ($\lambda_2 \rightarrow 0$), the deformation becomes unstable, leading to overfitting of local noise and visual artifacts, as indicated by higher LPIPS scores. Conversely, excessive regularization ($\lambda_2 \geq 0.1$) over-constrains the search space, preventing the model from correcting valid geometric errors and causing performance to drop towards

*Table 11.* **Ablation on regularization weight** $\lambda_2$ **(Deep Blending).** Extremely low values lead to artifacts, while high values limit detail recovery; 0.01 yields the best results.

| $\lambda_2$ | PSNR↑ | SSIM↑ | LPIPS↓ |
|---|---|---|---|
| 0 (w/o) | 29.80 | 0.895 | 0.250 |
| 0.001 | 30.10 | 0.905 | 0.240 |
| **0.01 (Default)** | **30.22** | **0.911** | **0.235** |
| 0.1 | 30.05 | 0.903 | 0.242 |
| 1.0 | 29.50 | 0.890 | 0.265 |

the baseline. We find that $\lambda_2 = 0.01$ yields the optimal balance, effectively anchoring the deformation to the proxy geometry to ensure consistency while allowing sufficient flexibility to retrieve high-frequency details.

## C. Architectural Analysis of the Geometry-Aware Deformable Offset

We justify choice of a lightweight MLP-based offset regressor over established dense correspondence networks through an analysis of computational bounds and geometric priors.

### C.1. Overcoming the Multi-View Computational Bottleneck

The aggregation of $M$ source views creates a strict computational budget for any feature refinement module. Conventional flow estimators (e.g., optical flow, DCN) are computationally expensive, often requiring heavy backbone networks to construct correlation volumes. Embedding such modules into our pipeline would require $M$ forward passes per pixel, leading to an explosion in GPU memory usage and training time. This overhead is incompatible with the real-time goals of 3D Gaussian Splatting. GADA circumvents this bottleneck by adopting a lightweight, coordinate-based MLP. This design decouples the complexity of deformation from the heavy feature extraction process, allowing for efficient, scalable correction across multiple views without compromising rendering speed.

### C.2. Leveraging Geometric Priors for Local Refinement

Unlike general optical flow tasks where correspondence must be established without prior knowledge, our system benefits from the explicit proxy geometry of 3D Gaussians. This proxy provides a coarse-level alignment, reducing the problem from global search to local residual correction. In this context, deep flow networks are over-parameterized and prone to overfitting or "texture copying" in the absence of strong constraints. Our approach treats the misalignment as a bounded local perturbation. By explicitly limiting the offset range, we align the network's capacity with the specific task of refining geometric projection errors, achieving optimal reconstruction quality with minimal model complexity.

# D. Additional Qualitative and Quantitative Results

In this section, we provide a detailed breakdown of our experimental results, including per-scene quantitative metrics and additional visual comparisons that could not be included in the main paper due to space constraints.

## D.1. Additional Qualitative Comparisons

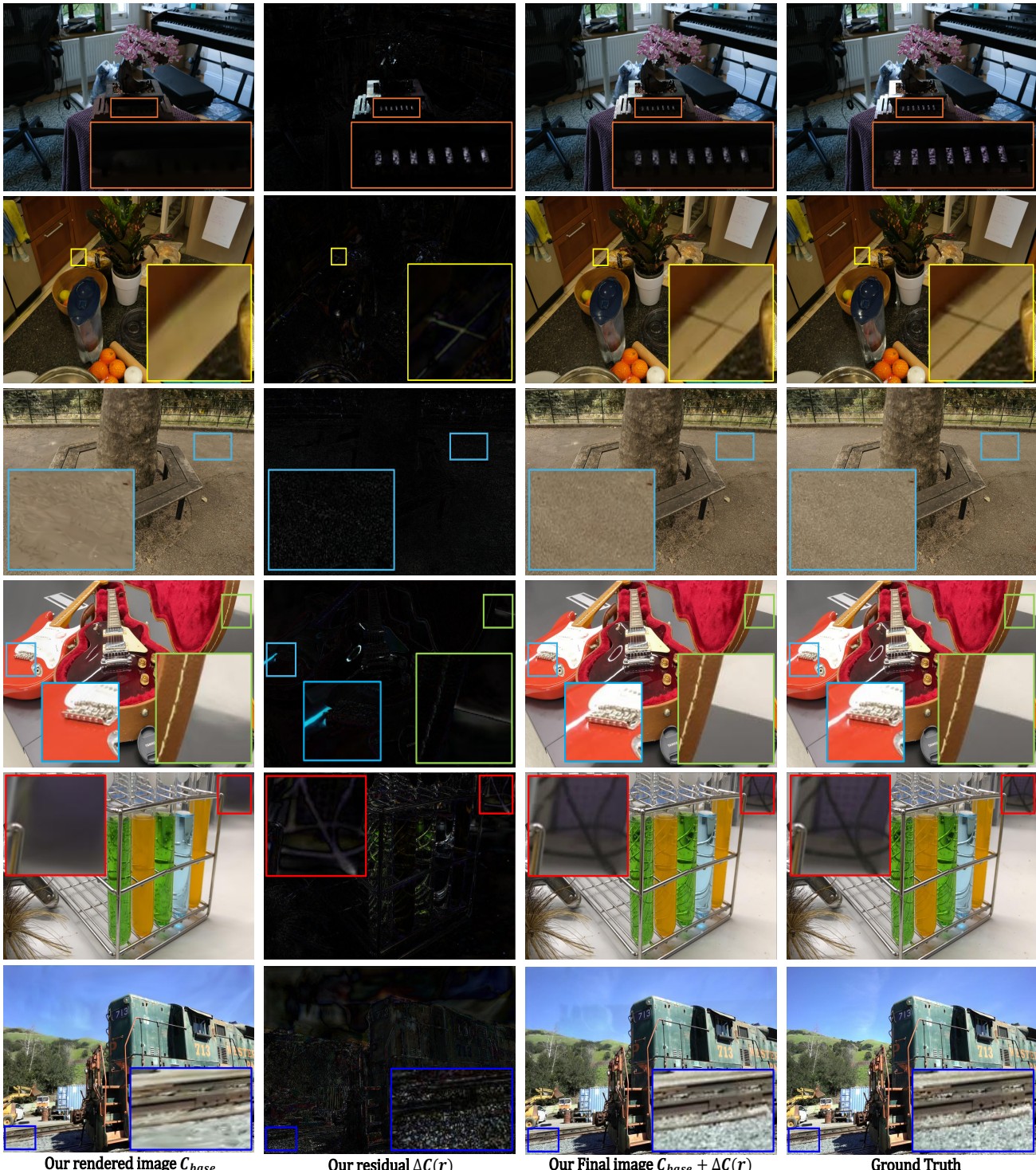

Our rendered image $C_{base}$     Our residual $\Delta C(r)$     Our Final image $C_{base} + \Delta C(r)$     Ground Truth

*Figure 10.* **Additional qualitative comparisons.** From left to right: base rendering ($C_{base}$), predicted residual ($\Delta C(r)$), final image, and GT. The insets highlight the residual's role in recovering missing high-frequency details, such as specular highlights and fine textures.

## D.2. Additional Quantitative Comparisons

*Table 12.* **Per-scene PSNR comparison on the Mip-NeRF 360 dataset.** We compare our method with state-of-the-art approaches. The best results are highlighted in **bold**.

| Method | Garden | Bicycle | Flowers | Treehill | Stump | Kitchen | Bonsai | Counter | Room |
|---|---|---|---|---|---|---|---|---|---|
| 2DGS | 26.69 | 24.77 | 21.14 | 22.36 | 26.20 | 30.41 | 31.30 | 28.10 | 30.37 |
| PGSR | 27.44 | 25.28 | 21.27 | 21.91 | 26.85 | 30.94 | 31.89 | 28.64 | 30.58 |
| Taming 3DGS | 27.42 | 24.75 | 21.10 | 23.02 | 25.95 | 30.92 | 31.86 | 28.53 | 31.41 |
| Octree-GS | 27.72 | 25.03 | 21.43 | 23.01 | 26.48 | 30.52 | 30.88 | 29.49 | 32.01 |
| 3DGS | 27.34 | 25.21 | 21.60 | 22.44 | 26.58 | 31.14 | 32.20 | 28.96 | 31.43 |
| Mip-Splatting | 27.47 | 25.25 | 21.60 | 22.65 | 26.64 | 31.25 | 31.96 | 29.04 | 31.54 |
| Scaffold-GS | 27.50 | 25.19 | 21.44 | 23.15 | 26.59 | 31.59 | 32.58 | 29.48 | 31.89 |
| AbsGS | 27.49 | 25.29 | 21.34 | 21.98 | 26.71 | 31.62 | 32.32 | 29.03 | 31.61 |
| 3DGS-MCMC | **27.81** | 25.69 | 22.05 | 22.97 | **27.38** | 31.91 | 32.66 | 29.32 | 32.05 |
| IBGS | 27.36 | 25.84 | 22.11 | 22.94 | 27.01 | 31.33 | 34.93 | 30.58 | 32.59 |
| **Ours** | 27.74 | **26.15** | **22.29** | **23.16** | 27.32 | **32.09** | **35.37** | **30.83** | **32.66** |

*Table 13.* **Per-scene SSIM comparison on the Mip-NeRF 360 dataset.** We compare our method with state-of-the-art approaches. The best results are highlighted in **bold**.

| Method | Garden | Bicycle | Flowers | Treehill | Stump | Kitchen | Bonsai | Counter | Room |
|---|---|---|---|---|---|---|---|---|---|
| 2DGS | 0.843 | 0.733 | 0.572 | 0.616 | 0.758 | 0.916 | 0.931 | 0.892 | 0.906 |
| PGSR | 0.870 | 0.780 | 0.618 | 0.623 | 0.787 | 0.923 | 0.941 | 0.910 | 0.917 |
| Taming 3DGS | 0.858 | 0.712 | 0.551 | 0.623 | 0.734 | 0.921 | 0.935 | 0.897 | 0.906 |
| Octree-GS | 0.869 | 0.759 | 0.600 | 0.643 | 0.763 | 0.917 | 0.923 | 0.907 | 0.922 |
| 3DGS | 0.866 | 0.764 | 0.604 | 0.631 | 0.771 | 0.926 | 0.941 | 0.907 | 0.917 |
| Mip-Splatting | 0.869 | 0.765 | 0.605 | 0.633 | 0.774 | 0.926 | 0.941 | 0.907 | 0.918 |
| Scaffold-GS | 0.863 | 0.759 | 0.592 | 0.640 | 0.766 | 0.927 | 0.943 | 0.910 | 0.922 |
| AbsGS | 0.871 | 0.783 | 0.623 | 0.617 | 0.780 | 0.929 | 0.945 | 0.911 | 0.925 |
| 3DGS-MCMC | **0.877** | 0.799 | 0.644 | 0.658 | **0.811** | **0.933** | 0.947 | 0.916 | 0.927 |
| IBGS | 0.862 | 0.791 | 0.652 | **0.676** | 0.767 | 0.918 | 0.958 | 0.926 | **0.934** |
| **Ours** | 0.864 | **0.805** | **0.657** | **0.676** | 0.809 | 0.929 | **0.959** | **0.927** | **0.934** |

*Table 14.* **Per-scene LPIPS comparison on the Mip-NeRF 360 dataset.** We compare our method with state-of-the-art approaches. The best results are highlighted in **bold**.

| Method | Garden | Bicycle | Flowers | Treehill | Stump | Kitchen | Bonsai | Counter | Room |
|---|---|---|---|---|---|---|---|---|---|
| 2DGS | 0.166 | 0.302 | 0.403 | 0.433 | 0.299 | 0.179 | 0.280 | 0.292 | 0.317 |
| PGSR | 0.116 | 0.204 | 0.304 | 0.322 | 0.225 | 0.161 | 0.244 | 0.246 | 0.277 |
| Taming 3DGS | 0.145 | 0.336 | 0.438 | 0.443 | 0.336 | 0.172 | 0.271 | 0.283 | 0.321 |
| Octree-GS | 0.124 | 0.251 | 0.374 | 0.360 | 0.276 | 0.172 | 0.280 | 0.262 | 0.275 |
| 3DGS | 0.124 | 0.240 | 0.367 | 0.377 | 0.250 | 0.155 | 0.254 | 0.258 | 0.287 |
| Mip-Splatting | 0.124 | 0.243 | 0.371 | 0.381 | 0.251 | 0.155 | 0.254 | 0.258 | 0.286 |
| Scaffold-GS | 0.136 | 0.259 | 0.382 | 0.373 | 0.277 | 0.156 | 0.249 | 0.256 | 0.275 |
| AbsGS | **0.100** | **0.171** | **0.270** | 0.278 | 0.195 | 0.121 | 0.190 | 0.189 | 0.200 |
| 3DGS-MCMC | 0.109 | 0.194 | 0.316 | 0.315 | 0.200 | 0.148 | 0.236 | 0.238 | 0.262 |
| IBGS | 0.136 | 0.192 | 0.286 | 0.285 | 0.199 | 0.132 | 0.156 | 0.156 | 0.177 |
| **Ours** | 0.123 | 0.172 | 0.277 | **0.272** | **0.183** | **0.114** | **0.149** | **0.151** | **0.174** |

*Table 15.* **Quantitative comparison on Deep Blending and Tanks & Temples datasets.** We report PSNR scores. The best results are highlighted in **bold**.

| Method | Deep Blending | | Tanks & Temples | |
|---|---|---|---|---|
| | Dr.Johnson | Playroom | Truck | Train |
| 2DGS | 28.75 | 30.23 | 25.10 | 21.17 |
| PGSR | 28.58 | 29.86 | 26.08 | 22.38 |
| Taming 3DGS | 29.51 | 30.24 | 25.86 | 22.23 |
| Octree-GS | **30.34** | 30.49 | 26.09 | 22.95 |
| 3DGS | 28.76 | 30.04 | 25.19 | 21.10 |
| Mip-Splatting | 28.71 | 30.01 | 25.71 | 21.78 |
| Scaffold-GS | 29.80 | **30.62** | 25.77 | 22.15 |
| AbsGS | 29.20 | 30.14 | 25.74 | 21.72 |
| 3DGS-MCMC | 29.00 | 30.33 | 26.11 | 22.47 |
| IBGS | 29.74 | 30.15 | 25.98 | 23.54 |
| **Ours** | 30.02 | 30.43 | **26.18** | **23.66** |

*Table 16.* **Quantitative comparison (SSIM) on Deep Blending and Tanks & Temples datasets.** Higher is better. The best results are highlighted in **bold**.

| Method | Deep Blending | | Tanks & Temples | |
|---|---|---|---|---|
| | Dr.Johnson | Playroom | Truck | Train |
| 2DGS | 0.900 | 0.907 | 0.873 | 0.793 |
| PGSR | 0.889 | 0.900 | 0.898 | 0.816 |
| Taming 3DGS | 0.906 | 0.909 | 0.892 | 0.811 |
| Octree-GS | **0.912** | **0.914** | **0.901** | 0.832 |
| 3DGS | 0.899 | 0.906 | 0.879 | 0.802 |
| Mip-Splatting | 0.898 | 0.907 | 0.893 | 0.826 |
| Scaffold-GS | 0.907 | 0.904 | 0.883 | 0.822 |
| AbsGS | 0.898 | 0.907 | 0.888 | 0.818 |
| 3DGS-MCMC | 0.890 | 0.900 | 0.890 | 0.830 |
| IBGS | 0.890 | 0.908 | 0.892 | 0.831 |
| **Ours** | 0.909 | 0.912 | 0.897 | **0.844** |

*Table 17.* **Quantitative comparison (LPIPS) on Deep Blending and Tanks & Temples datasets.** Lower is better. The best results are highlighted in **bold**.

| Method | Deep Blending | | Tanks & Temples | |
|---|---|---|---|---|
| | Dr.Johnson | Playroom | Truck | Train |
| 2DGS | 0.256 | 0.256 | 0.173 | 0.250 |
| PGSR | 0.254 | 0.257 | 0.157 | 0.177 |
| Taming 3DGS | 0.234 | 0.235 | 0.129 | 0.210 |
| Octree-GS | 0.241 | 0.236 | 0.129 | 0.178 |
| 3DGS | 0.244 | 0.241 | 0.148 | 0.218 |
| Mip-Splatting | 0.243 | 0.235 | 0.123 | 0.189 |
| Scaffold-GS | 0.250 | 0.258 | 0.147 | 0.206 |
| AbsGS | 0.240 | 0.232 | 0.131 | 0.193 |
| 3DGS-MCMC | 0.330 | 0.310 | 0.240 | **0.140** |
| IBGS | 0.232 | 0.240 | 0.127 | 0.182 |
| **Ours** | **0.231** | **0.239** | **0.116** | 0.173 |

## D.3. Additional efficiency Comparisons

*Table 18.* **Efficiency comparison on Mip-NeRF 360.** We measure training time, rendering speed (FPS), and total memory usage. Our method achieves a superior trade-off between quality and efficiency.

| Dataset | Mip-NeRF 360 | | |
|---|---|---|---|
| Method | Time(min)↓ | FPS(avg)↑ | Memory(MB)↓ |
| 3DGS | 30 | 196 | 764 |
| AbsGS | 29 | 132 | 728 |
| Scaffold-GS | 23 | 184 | 203 |
| Octree-GS | 28 | 106 | 419 |
| 3DGS-MCMC | 41 | 110 | 842 |
| IBGS | 43 | 22 | 295 |
| **Ours** | 52 | 47 | 275 |

*Table 19.* **Efficiency comparison on Deep Blending.** We measure training time, rendering speed (FPS), and total memory usage. Our method achieves a superior trade-off between quality and efficiency.

| Dataset | Deep Blending | | |
|---|---|---|---|
| Method | Time(min)↓ | FPS(avg)↑ | Memory(MB)↓ |
| 3DGS | 36 | 137 | 676 |
| AbsGS | 22 | 185 | 444 |
| Scaffold-GS | 24 | 139 | 66 |
| Octree-GS | 21 | 87 | 180 |
| 3DGS-MCMC | 30 | 138 | 750 |
| IBGS | 37 | 33 | 198 |
| **Ours** | 52 | 51 | 275 |

*Table 20.* **Efficiency comparison on Tanks&Temples** We measure training time, rendering speed (FPS), and total memory usage. Our method achieves a superior trade-off between quality and efficiency.

| Dataset | Tanks&Temples | | |
|---|---|---|---|
| Method | Time(min)↓ | FPS(avg)↑ | Memory(MB)↓ |
| 3DGS | 13 | 154 | 415 |
| AbsGS | 14 | 193 | 304 |
| Scaffold-GS | 16 | 110 | 87 |
| Octree-GS | 14 | 68 | 383 |
| 3DGS-MCMC | 21 | 126 | 474 |
| IBGS | 24 | 46 | 144 |
| **Ours** | 30 | 79 | 149 |

# E. Limitations and Discussion

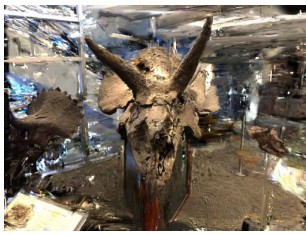 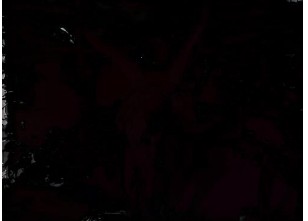

(a) Target view Rendering    (a) Residual Output

*Figure 11.* **Failure case under sparse view setting (LLHF Horn scene).** (a) Target View. (b) The predicted residual map provides no meaningful cues.

**Dependency on View Density (Sparse View Failure).** A fundamental limitation of our approach is its dependence on sufficient view overlap. Our method relies on aggregating warped features; however, when the angular distance between the target ray and the nearest source view is excessively large, occlusion and disocclusion phenomena become dominant. In such scenarios, the warped source images fail to contain valid photometric cues, leaving the network with no information to aggregate.

We empirically verify this limitation using the *Horn* scene from the LLFF dataset (Mildenhall et al., 2019) under a sparse-view setting. As shown in Fig. 11, due to the lack of proximal source views, the warped guidance is riddled with disocclusion holes (invalid regions). Consequently, the network effectively suppresses the residual prediction to avoid artifacts, failing to recover high-frequency details. This confirms that our geometry-aware aggregation requires a minimum level of view density to physically retrieve valid textures, mirroring the fundamental constraints of multi-view 3D reconstruction where performance degrades without overlapping visual information.

**Diffusion-assisted supervision for evidence-scarce regions.** The sparse-view failure case reveals a fundamental limitation of image-based reconstruction: GADA remains bounded by the availability of observed multi-view evidence. Although GADA can effectively correct locally misaligned warped cues and aggregate reliable source-view features, residual recovery becomes inherently underconstrained when severe occlusion, disocclusion, or insufficient view overlap removes valid photometric evidence from the source images.

This limitation suggests a possible extension beyond purely observed image-based cues. Recent advances in diffusion-based editing and fast generative modeling indicate that learned generative priors can provide useful structural guidance when direct visual evidence is sparse or ambiguous (Koo et al., 2024; Yoon et al., 2024; Labs et al., 2025; Koo et al., 2025; Podell et al., 2024). In this direction, diffu-

sion priors could be used not as a replacement for multi-view aggregation, but as an auxiliary source of weak supervision for regions where image-based evidence is unreliable.

A possible hybrid extension of GADA would therefore retain warped source-view evidence as the primary signal in well-observed regions, while selectively activating generative priors only in low-confidence regions identified by geometry-verified aggregation. Such a design could preserve the faithfulness and view consistency of image-based rendering, while providing controlled structural guidance for missing or severely underconstrained details.

