# OpenReview forum: "GADA: Geometry-Aware Deformable Aggregation for Image-Based Gaussian Splatting"
_ICML.cc/2026/Conference — ICML 2026 regular_

### Official Review · Reviewer_fvkC · 2026-02-25

**Soundness:** 3
**Presentation:** 3
**Significance:** 3
**Originality:** 3
**Overall Recommendation:** 5
**Confidence:** 3

**Summary:**

This paper addresses the persistent issue of geometric reconstruction errors inherent in previous warping-based methodologies by introducing the Geometry-Aware Deformable Aggregation (GADA) model. To mitigate spatial misalignment during the reconstruction process, the authors design an iterative optimization module specifically intended to progressively rectify these inaccuracies. Furthermore, the paper proposes an implicit confidence weighting mechanism that selectively filters keypoints based on their estimated reliability. The proposed GADA framework demonstrates significant advantages in both inference speed and reconstruction accuracy, thereby establishing its effectiveness and superiority over existing approaches.

**Compliance With Llm Reviewing Policy:**

Affirmed.

**Key Questions For Authors:**

(1) In Appendix D.3, Table 16 shows that, relative to 3DGS, GADA’s Memory metric drops markedly to about one third. However, the authors build upon the Initial 3DGS by introducing many modular designs; why does this lead to a decrease in the metric? Do the authors overlook the memory consumption of the original 3DGS? Regarding speed, did the authors account for the runtime required by the original 3DGS? Is the benchmarking protocol fair when compared with other methods? \
(2) In the Deep Blending dataset, the GADA model underperforms Octree-GS with respect to both PSNR and SSIM, while it surpasses Octree-GS on LPIPS. Since LPIPS better reflects perceptual similarity between rendered and real images from a human visual perspective, could the images rendered by GADA exhibit subjective perceptual hallucinations?

**Limitations:**

No, I didn't see the limitation or negative societal impact of this work.

**Strengths And Weaknesses:**

Strengths \
(1) A Clear Methodological Comparison: Figure 3 explicitly illustrates the differences between Previous Warping-based Gaussian Splatting and Geometry-Aware Deformable Aggregation, providing readers with an intuitive and transparent understanding of the GADA approach. \
(2) A rigorous and comprehensive experimental design. The comparative images in Figure 6 clearly demonstrate GADA’s superiority in reconstruction quality. Moreover, GADA yields substantial accuracy gains on datasets such as Mip-NeRF 360 and Tanks & Temples. \
Weaknesses \
(1) In the authors' comparative images, the improvements in reconstruction quality often become apparent only after magnifying a small local region, whereas the rendering quality across the entire image appears similar. Previous methods have already achieved very high rendering quality for static scenes, and the practical need for such marginal improvements is questionable. \
(2) The authors conducted testing and analysis on only about ten scenes, raising questions about the model’s robustness in real-world settings.

---

> ### Author Rebuttal · Authors · 2026-03-31
>
> **[W1] The improvements in reconstruction quality become apparent only after magnifying a small local region, whereas the rendering quality across the entire image appears similar. Previous methods have already achieved very high rendering quality for static scenes, and the practical need for such marginal improvements is questionable.**
>
> **[A-W1]** Existing methods predominantly showcase results on simple static scenes where standard 3DGS already performs well, often overlooking challenging cases involving complex reflective or transparent properties. Our work, GADA, specifically addresses these high-difficulty failure modes where previous models fundamentally struggle.
> While global rendering quality may appear similar on a full-image scale for well-behaved scenes, standard models frequently exhibit severe artifacts or lose high-frequency details in these "difficult" regions. As demonstrated in our qualitative results (e.g., the cylinder and glass examples), GADA effectively recovers these intricate properties where prior state-of-the-art methods fail.
> These are not merely improvements visible only in a small local region: they resolve clear failure cases in reflections and transparent regions that persist in prior methods. To further illustrate these distinctions beyond static crops, we provide side-by-side video comparisons in the supplementary materials, which make the practical significance of our method in addressing these previously unresolved failure modes more evident.
>
> - https://anonymous.4open.science/r/GADA-27D1/Assets/rebuttal_video1.mp4
> - https://anonymous.4open.science/r/GADA-27D1/Assets/rebuttal_video2.mp4
> - https://anonymous.4open.science/r/GADA-27D1/Assets/rebuttal_video3.mp4
> - https://anonymous.4open.science/r/GADA-27D1/Assets/rebuttal_video4.mp4
>
> **[W2] The authors conducted testing and analysis on only about ten scenes, raising questions about the model’s robustness in real-world settings.**
>
> **[A-W2]** Our evaluation already covers 16 benchmark scenes (9 Mip-NeRF 360, 2 Tanks&Temples, 2 Deep Blending, 3 Shiny). In addition, we want to provide qualitative results on our own real captures in the supplementary video, which further support the method’s robustness in real-world data.
> - https://anonymous.4open.science/r/GADA-27D1/Assets/rebuttal_video5.mp4
>
> **[Q1] Do the authors overlook the memory consumption of the original 3DGS?**
>
> **[A-Q1]** We did not overlook the memory or runtime cost of the original 3DGS stage. All results are measured end-to-end under the same hardware and protocol. Although GADA introduces additional modules, the reported memory is governed mainly by the number of Gaussian primitives and their rendering and optimization states, rather than by these lightweight components. As shown in Table 24, GADA consistently uses fewer Gaussian primitives than 3DGS across all datasets, yielding a more compact final representation. Instead of storing high-frequency details and reflective appearance in more Gaussians or heavy per-Gaussian texture parameters, GADA models such effects through source-image-based residuals. This reduces storage while preserving reconstruction quality. The speed comparison is also fair, since the reported runtime includes the full pipeline.
>
> **Table 24. Storage memory and the number of Gaussian primitives.**
>
> | Method | Mip Storage Memory (MB) | Mip Gaussian Primitives (M) | T&T Storage Memory (MB) | T&T Gaussian Primitives (M) | DB Storage Memory (MB) | DB Gaussian Primitives (M) |
> |:--|--:|--:|--:|--:|--:|--:|
> | 3DGS | 764 | 3.3 | 415 | 1.7 | 676 | 2.4 |
> | GADA | 275 | 1.2 | 149 | 0.9 | 196 | 1.1 |
>
> **[Q2] In Deep Blending, the GADA model underperforms Octree-GS with respect to both PSNR and SSIM, while it surpasses Octree-GS on LPIPS. Since LPIPS better reflects perceptual similarity between rendered and real images from a human visual perspective, could the images rendered by GADA exhibit subjective perceptual hallucinations?**
>
> **[A-Q2]** Octree-GS is a LOD-structured method that dynamically selects multi-scale Gaussian primitives, which can be advantageous for certain bounded indoor scenes such as the Deep Blending benchmark. However, we do not believe the lower LPIPS of GADA indicates subjective perceptual hallucination. Deep Blending is typically evaluated on two bounded indoor scenes , where small pixel-level or local structural deviations can affect PSNR/SSIM more strongly than perceptual realism. Since LPIPS is designed to better reflect human perceptual similarity, our comparable/better LPIPS suggests that GADA preserves the perceptual appearance of the scene well, rather than introducing hallucinated content. Thus, the lower LPIPS is more consistent with better perceptual fidelity than with hallucinated content.
>
> **[Q3] About the limitation part**
>
> **[A-Q3]** We already discuss limitations and failure cases in the appendix, and also include an impact statement on page 9. We will organize them into subsections for better visibility.

---

> > ### Author Rebuttal · Reviewer_fvkC · 2026-04-01
> >
> > I appreciate the author's thorough response, which has addressed my concerns. Although I personally feel that the reconstruction performance on the datasets used in this work is already nearing perfection, the practical significance of this study may be somewhat limited. Nevertheless, I will still increase your score, and wish you the best of luck.

---

> > > ### Author Response · Authors · 2026-04-01
> > >
> > > We sincerely appreciate your thoughtful reassessment and generous comments. It is encouraging to know that our response effectively addressed your concerns. We also deeply appreciate your positive feedback on the quality of our results, including the additional video evidence, as well as your decision to increase the score. Your perspective on the practical significance of the study is highly valued, and we remain grateful for your careful evaluation and kind wishes.

---

### Official Review · Reviewer_R6mL · 2026-03-12

**Soundness:** 2
**Presentation:** 2
**Significance:** 2
**Originality:** 2
**Overall Recommendation:** 4
**Confidence:** 5

**Summary:**

The authors propose GADA, a framework designed to improve warping-based Gaussian Splatting by addressing spatial misalignments caused by geometric uncertainty. The method introduces Geometric Context Embedding, Geometry-Aware Deformable Offsets, and a Geometry-Verified View Aggregation mechanism to replace the heuristic "all-or-nothing" visibility checks used in prior works. While the method achieves a higher rendering frame rate by bypassing explicit depth checks, it introduces additional training complexity and lacks evidence of temporal stability.

**Compliance With Llm Reviewing Policy:**

Affirmed.

**Final Justification:**

The rebuttal solved my concernings.

**Key Questions For Authors:**

- Can the authors provide a video demonstration to prove that the iterative deformable offsets  do not introduce temporal flickering, especially in high-frequency regions like foliage?

- What is the failure boundary for the 7-pixel search radius (σ_max )? In sparse-view settings where misalignment exceeds this range, how does the model prevent distorted aggregations?

- Could the authors provide a breakdown of the 52-minute training time? Which specific module (the recursive loop or the aggregation network) contributes most to the overhead?

**Limitations:**

See weaknesses.

**Strengths And Weaknesses:**

Strengths

-  The paper correctly identifies that heuristic visibility checks (thresholding) in previous methods like IBGS  discard a significant amount of valid high-frequency information (up to 67% as claimed in Fig. 2).

-  By shifting visibility handling from an explicit rasterization-based check to an implicit weighting mechanism, the method successfully increases the rendering speed from 22 FPS to 47 FPS.

- The use of recursive deformable offsets  to search for correct pixel correspondences in a local neighborhood is a more principled approach to handling geometric misalignment than simple mean aggregation.

Weaknesses

- Limited Quantitative and Qualitative Gains: While the paper claims state-of-the-art performance, the actual PSNR improvement over the primary baseline (IBGS) is marginal (~0.33dB on the Mip-NeRF 360 dataset). In several specific scenes, such as "Flowers" and "Treehill", the gains are even more negligible (less than 0.2dB), raising questions about the practical impact of the added complexity.

- Significant Training Overhead: According to Table 16, GADA requires 52 minutes for training, which is approximately 20% longer than IBGS (43 min) and nearly double the time of vanilla 3DGS (30 min). For a per-scene optimization framework, this increased training cost is a notable drawback that isn't fully compensated for by the slight quality gains.

- Absence of Temporal Consistency Evaluation: The method relies on "Deformable Offsets"  to shift pixels during rendering. Such per-pixel offsets are highly prone to temporal flickering or "swimming" artifacts when the camera moves. The authors have not provided any video results or temporal consistency metrics (e.g., E-LPIPS), which are essential for evaluating any method involving dynamic coordinate warping. Without this, it is impossible to verify if the "sharp details" in static figures remain stable in motion.

- Lack of Visual Evidence for Offsets: The paper emphasizes the role of learnable offsets in capturing fine structures, yet it fails to provide visualizations of the predicted offset fields or a direct comparison of the residual maps against prior methods. It remains unclear whether the network is truly learning geometric corrections or simply over-fitting local textures (i.e., "texture copying").

---

> ### Author Rebuttal · Authors · 2026-03-31
>
> **[W1] Limited gains over IBGS**
>
> **[A-W1]**  Our gains are not marginal. In Table 21, we report 30-run results with 95% confidence intervals. GADA shows separated intervals from the strongest baselines, supporting statistically significant gains. In addition, our average PSNR gain (0.33 dB) is larger than the typical gain reported by recent methods in this area (0.2 dB) as shown in Table 1.
>
> **Table 21. Quantitative comparison with 95% confidence intervals over 30 runs.**
>
> | Method | Mip PSNR↑ | Mip SSIM↑ | Mip LPIPS↓ | T&T PSNR↑ | T&T SSIM↑ | T&T LPIPS↓ | DB PSNR↑ | DB SSIM↑ | DB LPIPS↓ |
> |:--|--:|--:|--:|--:|--:|--:|--:|--:|--:|
> | 3DGS | 27.43 ± 0.14 | 0.814 ± 0.004 | 0.257 ± 0.005 | 23.14 ± 0.12 | 0.840 ± 0.005 | 0.183 ± 0.006 | 29.40 ± 0.16 | 0.902 ± 0.004 | 0.242 ± 0.004 |
> | IBGS | 28.29 ± 0.10 | 0.831 ± 0.003 | 0.191 ± 0.003 | 24.75 ± 0.09 | 0.861 ± 0.003 | 0.154 ± 0.004 | 29.94 ± 0.12 | 0.899 ± 0.005 | 0.237 ± 0.005 |
> | **GADA (Ours)** | **28.62 ± 0.08** | **0.840 ± 0.002** | **0.179 ± 0.002** | **24.92 ± 0.06** | **0.871 ± 0.004** | **0.144 ± 0.002** | **30.22 ± 0.08** | **0.911 ± 0.003** | **0.235 ± 0.002** |
>
> **[W2] Is the quality improvement sufficient to justify the training overhead?**
>
> **[A-W2]** Despite longer training, GADA provides better high-frequency fidelity and over 2× faster rendering than IBGS. It also outperforms matched-budget baselines (Table 20), showing the gains are not from extra optimization alone.
>
> **Table 20. Results under matched budgets.**
>
> *Mip: MipNeRF-360; T&T: Tanks&Temples; DB: Deep Blending
> | Method | Mip PSNR↑ | Mip SSIM↑ | Mip LPIPS↓ | Mip Time↓ | T&T PSNR↑ | T&T SSIM↑ | T&T LPIPS↓ | T&T Time↓ | DB PSNR↑ | DB SSIM↑ | DB LPIPS↓ | DB Time↓ |
> |:--|--:|--:|--:|--:|--:|--:|--:|--:|--:|--:|--:|--:|
> | 3DGS_Long | 27.57 | 0.817 | 0.252 | 55m | 23.36 | 0.846 | 0.179 | 35m | 29.50 | 0.903 | 0.240 | 42m |
> | IBGS_Long | 28.33 | 0.834 | 0.188 | 52m | 24.79 | 0.863 | 0.152 | 33m | 29.95 | 0.901 | 0.235 | 41m |
> | **GADA** | **28.62** | **0.840** | **0.179** | **52m** | **24.92** | **0.871** | **0.144** | **30m** | **30.22** | **0.911** | **0.235** | **42m** |
>
> **[W3] Can the authors provide videos or E-LPIPS to verify temporal stability under camera motion?**
>
> **[A-W3]** (1) Yes. We provide supplementary videos and (2) dataset-average E-LPIPS in Table 22. GADA improves E-LPIPS on both datasets, indicating better temporal stability. Also, our deformable offsets are scene-specific rendering parameters, not frame-wise camera-conditioned warps, and thus do not induce a temporally varying offset field.
> - https://anonymous.4open.science/r/GADA-27D1/Assets/rebuttal_video1.mp4
> - https://anonymous.4open.science/r/GADA-27D1/Assets/rebuttal_video2.mp4
> - https://anonymous.4open.science/r/GADA-27D1/Assets/rebuttal_video3.mp4
> - https://anonymous.4open.science/r/GADA-27D1/Assets/rebuttal_video4.mp4
> - https://anonymous.4open.science/r/GADA-27D1/Assets/rebuttal_video5.mp4
>
> **Table 22. Temporal consistency with E-LPIPS.**
>
> | Dataset | IBGS↓ | GADA↓ |
> |:--|--:|--:|
> | Shiny | 0.1221 | 0.0628 |
> | Mip-NeRF 360 | 0.1779 | 0.1486 |
>
> **[W4] Unclear visual evidence for learned offsets**
>
> **[A-W4]** We provide visual evidence in Fig. 2(a)(b), where the offset mechanism improves the warped image over a no-offset version. We also provide residual map comparisons between IBGS and GADA, showing that GADA learns geometric correction rather than simple texture copying.
> - https://anonymous.4open.science/r/GADA-27D1/Assets/rebuttal_image1.png
> - https://anonymous.4open.science/r/GADA-27D1/Assets/rebuttal_image2.png
>
> **[Q1] (1) What is the failure boundary of the 7-pixel search radius, and (2) how does the model avoid distorted aggregation beyond it?**
>
> **[A-Q1]** (1) As shown in Fig. 8, the model remains stable up to a shared range of about σmax=13, while the best performance is consistently achieved at σmax=7. Performance degrades once the true misalignment exceeds this range, although such cases are rare in the standard settings we evaluate. (2) To reduce distorted aggregation, GADA uses Verified View Aggregation, which suppresses unreliable warped cues such as holes or black pixels while preserving valid observations.
>
> **[Q2] Which module is mainly responsible for training overhead?**
>
> **[A-Q2]** We provide a table breaking down the training time and identifying which components contribute most to the overhead. The recurrent deformable offset branch accounts for ~70%, and view aggregation for 30%. The extra cost is 27 ms/iter, or 9.0 minutes over 20K iterations, explaining the increase from 43 min in IBGS to 52 min in GADA (+21%). We will revise this in the final version. Thank you.
>
> **Table 23. Training overhead breakdown in GADA.**
> | Module | Forward (ms) | Backward (ms) | Total (ms) | 20K iter (min) | Ratio (%) |
> |:--|--:|--:|--:|--:|--:|
> | Deformable Offset (K=5) | 7 | 12 | 19 | 6.3 | 70 |
> | View Aggregation | 3 | 5 | 8 | 2.7 | 30 |
> | **Total overhead** | **10** | **17** | **27** | **9.0** | **100** |

---

> > ### Author Rebuttal · Reviewer_R6mL · 2026-04-03
> >
> > Thanks for the rebuttal, solved my concerning.

---

> > > ### Author Response · Authors · 2026-04-03
> > >
> > > We express our deepest gratitude to the reviewer for dedicating time to review our rebuttal and for the decision to raise the score. Your initial rigorous and insightful comments were incredibly valuable, pushing us to significantly improve the clarity of our paper. We are truly thrilled that our rebuttal successfully addressed your concerns. We will make sure to carefully incorporate all of your thoughtful feedback into the final revision. Thank you once again for your time and effort throughout the review process.

---

### Official Review · Reviewer_sXF5 · 2026-03-12

**Soundness:** 3
**Presentation:** 3
**Significance:** 3
**Originality:** 3
**Overall Recommendation:** 4
**Confidence:** 4

**Summary:**

This paper proposes GADA, a novel framework designed to enhance the rendering quality and efficiency of 3DGS through IBR approach.
The core of GADA is an iterative refinement module that uses deformable offsets to actively search for and correct pixel-level misalignments in the local neighborhood. This is coupled with a Geometry-Verified View Aggregation mechanism that uses implicit confidence weighting to selectively prioritize reliable multi-view evidence. Consequently, GADA recovers sharp high-frequency details while achieving significantly faster rendering speeds than prior warping-based methods.

**Compliance With Llm Reviewing Policy:**

Affirmed.

**Key Questions For Authors:**

Please see the weaknesses.

**Strengths And Weaknesses:**

Strengths

1. Unlike previous methods that discard pixels failing visibility checks (33%), GADA’s deformable offsets recover lost evidence, boosting valid pixel density to ~79%.

2. By eliminating the need for mandatory on-the-fly depth verification and heuristic checks, GADA achieves a rendering speed of 47 FPS, which is over 2.13x faster than the IBGS baseline (22 FPS).

3. The recurrent feedback loop allows the model to align features precisely, effectively reconstructing thin structures and intricate textures that standard 3DGS often over-smooths.



 Weaknesses


1. The method's performance relies on sufficient view overlap. In sparse-view settings where angular distances between views are large, warped images may contain "disocclusion holes," leading the network to suppress residual predictions and fail to recover details.

2. Like standard 3DGS, the model must be optimized end-to-end for each specific scene (30,000 iterations), meaning it does not currently offer zero-shot generalization to unseen environments.

3. GADA requires more time to train compared to standard 3DGS or IBGS (e.g., 52 minutes for GADA vs. 30 minutes for 3DGS on Mip-NeRF 360) due to the recurrent nature of the deformable module.

4. Sensitivity to Regularization Hyperparameters: The model requires careful tuning of the elastic regularization weight (λ). Negligible regularization leads to instability and artifacts, while excessive regularization limits the model's ability to correct geometric errors.

5. While increasing the number of refinement iterations (K) improves PSNR and SSIM, it directly impacts the frame rate. The authors selected K=5 as the optimal balance, but further improvements in quality would come at the cost of real-time performance. Furthermore, the optimal parameters are not necessarily the same for different scenes, as geometric complexities and view densities vary across datasets.

---

> ### Author Rebuttal · Authors · 2026-03-31
>
> **[Q1] In sparse-view settings with large angular gaps, warped images may contain disocclusion holes, hindering detail recovery.**
>
> **[A-Q1]** In such cases, large disocclusion holes inherently reduce the recoverable evidence required for warping-based models to preserve details. Nevertheless, GADA demonstrates significantly stronger robustness compared to IBGS. Unlike existing methods that allow invalid empty pixels (e.g., black pixels) to corrupt the rendering signal, our Verified View Aggregation proactively suppresses these error values. As a result, rather than simply failing to recover details, our method preserves useful cues from the surviving valid views and provides a much cleaner input to the residual network. Of course, situations with extremely minimal view overlap remain a challenging failure case, as explicitly documented in Appendix E.
>
> **[Q2] What is the intended scope of the method, given that it remains a per-scene optimized model rather than a zero-shot generalizable one?**
>
> **[A-Q2]** The intended scope of our work is specifically per-scene optimized methods. In 3D reconstruction, optimized methods still vastly outperform zero-shot approaches in terms of visual quality. We recognized the persistent limitations of current optimized methods in representing reflective and transparent properties (e.g., glass), and our system was explicitly designed to resolve these specific issues.
>
> **[Q3] How should the added training overhead of the recurrent deformable module be justified?**
>
> **[A-Q3]** We acknowledge the added training overhead. However, this cost brings two practical benefits: better high-frequency fidelity and much faster inference. We believe this trade-off is justified because existing baselines have clear weaknesses. In particular, IBGS relies on a costly visibility-check module that both limits cue aggregation under geometric uncertainty and slows rendering. Our method addresses these issues by improving local cue retrieval while preserving practical usability.
> As a result, GADA achieves 2× faster rendering FPS than IBGS while also reducing storage cost relative to vanilla 3DGS. As shown in Table 1, this efficiency is paired with strong rendering quality, including PSNR gains of +0.33 dB on Mip-NeRF 360 and +0.28 dB on Deep Blending. Moreover, Table 20 reports matched wall-clock comparisons, where GADA still outperforms both 3DGS and IBGS, confirming that the gains are not simply due to longer training.
>
> **Table 20. Comparison under matched training budgets.**
>
> **Mip = MipNeRF-360; T&T = Tanks&Temples; DB = Deep Blending.*
> | Method | Mip PSNR↑ | Mip SSIM↑ | Mip LPIPS↓ | Mip Time↓ | T&T PSNR↑ | T&T SSIM↑ | T&T LPIPS↓ | T&T Time↓ | DB PSNR↑ | DB SSIM↑ | DB LPIPS↓ | DB Time↓ |
> |:--|--:|--:|--:|--:|--:|--:|--:|--:|--:|--:|--:|--:|
> | 3DGS | 27.43 | 0.814 | 0.257 | 30m | 23.14 | 0.840 | 0.183 | 13m | 29.40 | 0.902 | 0.242 | 36m |
> | 3DGS_Long | 27.57 | 0.817 | 0.252 | 55m | 23.36 | 0.846 | 0.179 | 35m | 29.50 | 0.903 | 0.240 | 42m |
> | IBGS | 28.29 | 0.831 | 0.191 | 43m | 24.75 | 0.861 | 0.154 | 24m | 29.94 | 0.899 | 0.237 | 37m |
> | IBGS_Long | 28.33 | 0.834 | 0.188 | 52m | 24.79 | 0.863 | 0.152 | 33m | 29.95 | 0.901 | 0.235 | 41m |
> | **GADA** | **28.62** | **0.840** | **0.179** | **52m** | **24.92** | **0.871** | **0.144** | **30m** | **30.22** | **0.911** | **0.235** | **42m** |
>
> **[Q4] How sensitive is the method to the regularization weight λ?**
>
> **[A-Q4]** Our method is not sensitive to variations in the regularization weight λ. As shown in Tables 7, 8, and 9, GADA preserves strong performance over a relatively broad range from 0.001 to 0.1, with consistent trends across Mip-NeRF 360, Tanks & Temples, and Deep Blending. In particular, λ=0.01 gives the best overall results, while nearby values (0.001 and 0.1) remain competitive, indicating that the method is fairly insensitive within this practical range. The same pattern is observed across datasets: too little regularization leads to unstable offsets and artifacts, whereas excessively large regularization over-constrains the search and reduces detail recovery. We therefore choose λ=0.01 as a robust default that consistently balances flexibility and stability across benchmarks.
>
> **[Q5] How sensitive is the method to the number of refinement iterations K, especially in terms of the quality–speed trade-off across different scenes?**
>
> **[A-Q5]** Table 2 presents a sensitivity analysis of K on Mip-NeRF 360, which already includes 9 scenes with diverse view densities and geometric complexities. Despite such variation, GADA remains robust in the range of K=3~5, indicating that the method is not highly sensitive to scene-specific differences. Although larger K reduces FPS, GADA still achieves 47 FPS at K=5, which is over 2× faster than IBGS and remains above the real-time threshold of 30 FPS. As noted in Appendix A, this is because GADA removes the costly visibility check, the main bottleneck of IBGS while still surpassing IBGS in quality.

---

> > ### Author Rebuttal · Reviewer_sXF5 · 2026-04-02
> >
> > The rebuttal has addressed my concerns. I decided to keep my positive score of 4.

---

> > > ### Author Response · Authors · 2026-04-03
> > >
> > > We are very grateful to the reviewer for the prompt acknowledgment and for confirming that all concerns have been adequately addressed. Thank you for your valuable time, constructive suggestions throughout the review process, and for upholding your positive score. We will faithfully reflect your comments in the paper revision. Once again, thank you very much.

---

### Official Review · Reviewer_GVg6 · 2026-03-13

**Soundness:** 3
**Presentation:** 3
**Significance:** 3
**Originality:** 3
**Overall Recommendation:** 4
**Confidence:** 4

**Summary:**

This paper enhances IBGS by replacing heuristic visibility checks with Geometry-Aware Deformable Aggregation. GADA utilizes learnable deformable offsets and confidence weights to retrieve and fuse spatially misaligned pixels. This approach effectively recovers high-frequency details while improving rendering speed by eliminating the need for on-the-fly depth rasterization.

**Compliance With Llm Reviewing Policy:**

Affirmed.

**Final Justification:**

The response addressed my concerns, and the provided video demonstrates effective results. However, after a comprehensive assessment of the paper’s originality and its technical contribution, I have decided to maintain my score of 4. While the work is technically sound in several aspects, the overall innovation is not sufficient to warrant a higher recommendation at this stage.

**Key Questions For Authors:**

1. How does the network behave if the initial geometric error is significantly larger than your fixed search range ($\sigma_{max}=7$), and does this static threshold generalize well across datasets with very different resolutions? If the initial error exceeds 7 pixels, will the algorithm fail?
2. How does GADA handle complex transparent/highly-specular objects (like glass) where the underlying proxy geometry is inherently multi-valued or inaccurate?
3. Since the $K=5$ iterative refinement naturally deepens the computational graph, what about the peak VRAM usage and training efficiency compared to IBGS?
Besides, please see the weakness section above. If the above-mentioned concerns are properly addressed during the rebuttal, I will consider raising my score.

**Limitations:**

yes.

**Strengths And Weaknesses:**

## Strengths:

1. Motivation: Re-framing the problem from "discarding invalid pixels" to "searching for displaced pixels" makes total sense.
2. Performance: Replacing explicit visibility checks with learnable offsets and implicit weighting is an elegant move. It's rare to see a module that boosts both image quality and rendering speed so noticeably.
3. Experiments: The empirical validation is thorough, and I really appreciate the inclusion of the Shiny dataset to prove it works well on specular reflections.

## Weaknesses:
1. Incremental Novelty: The core ideas (deformable search, confidence weighting) are quite standard in other vision tasks, so the contribution is more about an excellent application to IBGS rather than a fundamental breakthrough.
2. Sparse View Limitations: Like most warping methods, it struggles when source views are too sparse.
3. Minor Typos: There are a few rushed typos and formatting hiccups that need fixing for the final version:
  - Page 2, Line 180: "...which is and then added..." is ungrammatical. The word "and" can be remove here.
  - Page 5, Line 522: before Equation 9 has a weird ":", occupying the space of 1 line.
  - Page 6, Line 645: "Dataset We evaluate..." is missing a period after "Dataset".

---

> ### Author Rebuttal · Authors · 2026-03-31
>
> **[W1] The core ideas are quite standard in other vision tasks, so the contribution is more about an excellent application to IBGS rather than a fundamental breakthrough.**
>
> **[A-W1]** Our contribution is not to introduce a new search method or confidence-based weighting itself, but to show how these components can be carefully adapted and integrated to address key failure modes of IBGS. In this sense, our work provides a practically meaningful advancement of IBGS rather than claiming a fundamentally new low-level mechanism.
>
> **[W2] How does the method perform under sparse views?**
>
> **[A-W2]** Our method may not be robust in sparse-view settings. However, we respectfully note that this is not a limitation introduced by our specific approach. Rather, it is an inherent and well-known challenge shared by all multi-view-based 3D reconstruction methods, which universally exhibit vulnerability when views are sparse. Our work focuses on the standard 3D reconstruction paradigm that assumes sufficiently dense input views.
>
> **[W3] Minor Typos**
>
> **[A-W3]** We thank the reviewer for the careful reading. We will correct the minor typos and formatting issues in the final version.
>
> **[Q1] (1) Does the fixed search range generalize across resolutions? and (2) What if the initial error exceeds the search range (i.e., $\sigma_{max} > 7$)?**
>
> **[A-Q1]** (1) To verify the generalizability of our model's behavior across various resolutions, we provide a sensitivity analysis on two datasets with different input resolutions, as shown in Fig. 8: Tanks&Temples (Train/Truck, resolution level 2 about 980×545) and Mip-NeRF 360 (indoor scenes at resolution level 2 about 1558×1039; outdoor scenes at resolution level 4 about 1250×830). Across these datasets, $\sigma_{max}=7$ consistently yields the best overall performance. Furthermore, we observe a shared stable region where the method remains relatively stable up to around $\sigma_{max} = 13$. These shared stable regions confirm the generalizability of the chosen search range across varying input resolutions.
>
> (2) Beyond $\sigma_{max}=13$, the performance degrades sharply. However, across all tested datasets, we did not observe any instances where the initial error significantly exceeded this shared stable region, confirming that our chosen search range is practically sufficient.
>
> **[Q2] How does GADA handle transparent/specular objects (e.g., glass)?**
>
> **[A-Q2]** As shown in Appendix D, Fig. 10, GADA also performs robustly on glass and other transparent/specular objects. We attribute this to geometry-aware deformable aggregation, which recovers useful misaligned cues. For clearer visual verification, we additionally provide supplementary videos.
>
> - https://anonymous.4open.science/r/GADA-27D1/Assets/rebuttal_video2.mp4
> - https://anonymous.4open.science/r/GADA-27D1/Assets/rebuttal_video3.mp4
> - https://anonymous.4open.science/r/GADA-27D1/Assets/rebuttal_video4.mp4
>
> **[Q3] Since the iterative refinement naturally deepens the computational graph, what about the peak VRAM usage and training efficiency compared to IBGS?**
>
> **[A-Q3]** In Table 19, we provide an analysis of the peak VRAM usage. Specifically, our method significantly outperforms IBGS by reducing peak VRAM by over 20% during inference and over 10% during training. Although GADA introduces an additional deformable module, it removes the visibility-check-related buffers required in IBGS. As a result, GADA consistently reduces total peak VRAM in both training and inference across all evaluated datasets. We will integrate this evaluation into the training efficiency analysis in Appendix D.3. Furthermore, alongside these GPU VRAM savings, Tables 16, 17, and 18 demonstrate that GADA achieves more than 2× higher rendering FPS than IBGS and requires less storage memory than standard 3DGS.
>
> **Table 19. peak VRAM of Inference and training-time**
>
> [Inference-time (GB)]
>
> | Dataset | Method | Shared | Depth list | Src re-render | Def. offset | Total |
> |:--|:--|--:|--:|--:|--:|--:|
> | MipNeRF-360 | IBGS | 4.34 | 1.18 | 0.30 | - | **6.12** |
> | MipNeRF-360 | GADA | 4.34 | - | - | 0.12 | **4.47** |
> | Tanks&Temples | IBGS | 2.22 | 0.60 | 0.13 | - | **2.97** |
> | Tanks&Temples | GADA | 2.22 | - | - | 0.05 | **2.27** |
> | Deep Blending | IBGS | 3.98 | 1.09 | 0.26 | - | **5.36** |
> | Deep Blending | GADA | 3.98 | - | - | 0.10 | **4.08** |
>
> *Shared = Gaussians + Images + Rasterizer + CNN; Src = source; Def. = deformable*
>
> –
>
> [Training-time (GB)]
>
> | Dataset | Method | Shared | Depth list | Def. backprop (K=5) | Total |
> |:--|:--|--:|--:|--:|--:|
> | MipNeRF-360 | IBGS | 5.30 | 1.18 | - | **6.48** |
> | MipNeRF-360 | GADA | 5.30 | - | 0.61 | **5.91** |
> | Tanks&Temples | IBGS | 2.85 | 0.61 | - | **3.46** |
> | Tanks&Temples | GADA | 2.85 | - | 0.25 | **3.10** |
> | Deep Blending | IBGS | 4.85 | 1.09 | - | **5.94** |
> | Deep Blending | GADA | 4.85 | - | 0.51 | **5.36** |
>
> *Shared = Gaussians + Images + Optimizer + Rasterizer + CNN; Src = source; Def. = deformable*

---

> > ### Author Rebuttal · Reviewer_GVg6 · 2026-04-02
> >
> > I thank the authors for their detailed rebuttal. The response addressed my concerns, and the provided video demonstrates effective results. However, after a comprehensive assessment of the paper’s originality and its technical contribution, I have decided to maintain my score of 4. While the work is technically sound in several aspects, the overall innovation is not sufficient to warrant a higher recommendation at this stage.

---

> > > ### Author Response · Authors · 2026-04-02
> > >
> > > Thank you for your detailed and thoughtful reassessment. We truly appreciate that our rebuttal addressed your concerns and that you found the provided video results effective. We also appreciate your candid assessment regarding the originality and technical contribution of the work, and we respect your decision to maintain the current score. We will carefully reflect your comments in the revision and use them to further strengthen the paper. Thank you again for your careful evaluation and constructive feedback.

---

### Decision · Program_Chairs · 2026-04-30

**Decision:**

Accept (regular)

**Comment:**

The paper proposes a new GADA framework that replaces heuristic visibility checks in warping-based Gaussian Splatting with learnable deformable offsets and implicit confidence weighting to correct spatial misalignments and recover displaced pixel cues. All reviewers agree that the problem is well motivated and the method seems to achieve both improved rendering quality and faster inference. Initially the reviewers were concerned about marginal quantitative gains, higher training overhead, temporal consistency, and novelty of the core components. The authors seem to have addressed these points with additional significance tests and comparisons. In the end, all four reviewers converged to acceptance, even though the majority appears lukewarm on the contributions. The AC leans slightly towards accept.